# Overcalcified forms of the coccolithophore *Emiliania huxleyi* in high CO₂ waters are not pre-adapted to ocean acidification.

Peter von Dassow[1,2,3*], Francisco Díaz-Rosas[1,2], El Mahdi Bendif[4], Juan-Diego Gaitán-Espitia[5], Daniella Mella-Flores[1], Sebastian Rokitta[6], Uwe John[6,7], and Rodrigo Torres[8,9]

[1] Facultad de Ciencias Biológicas, Pontificia Universidad Católica de Chile, Santiago, Chile.
[2] Instituto Milenio de Oceanografía de Chile.
[3] UMI 3614, Evolutionary Biology and Ecology of Algae, CNRS-UPMC Sorbonne Universités, PUCCh, UACH.
[4] Department of Plant Sciences, University of Oxford, OX1 3RB Oxford, UK.
[5] CSIRO Oceans and Atmosphere, GPO Box 1538, Hobart 7001, TAS, Australia.
[6] Alfred Wegener Institute – Helmholtz Centre for Polar and Marine Research, Bremerhaven, Germany.
[7] Helmholtz Institute for Functional Marine Biodiversity (HIFMB), Ammerländer Heerstr. 231, 26129 Oldenburg, Germany.
[7] Centro de Investigación en Ecosistemas de la Patagonia (CIEP), Coyhaique, Chile.
[8] Centro de Investigación: Dinámica de Ecosistemas marinos de Altas Latitudes (IDEAL), Punta Arenas, Chile.

*Correspondence to*: Peter von Dassow (pvondassow@bio.puc.cl)

**Abstract.** Marine multicellular organisms inhabiting waters with natural high fluctuations in pH appear more tolerant to acidification than conspecifics occurring in nearby stable waters, suggesting that environments of fluctuating pH hold genetic reservoirs for adaptation of key groups to ocean acidification (OA). The abundant and cosmopolitan calcifying phytoplankton *Emiliania huxleyi* exhibits a range of morphotypes with varying degrees of coccolith mineralization. We show that *E. huxleyi* populations in the naturally acidified upwelling waters of the Eastern South Pacific, where pH drops below 7.8 as is predicted for the global surface ocean by the year 2100, are dominated by exceptionally overcalcified morphotypes whose distal coccolith shield can be almost solid calcite. Shifts in morphotype composition of *E. huxleyi* populations correlate with changes in carbonate system parameters. We tested if these correlations indicate that the hypercalcified morphotype is adapted to OA. In experimental exposures to present-day vs. future $p$CO₂ (400 µatm vs. 1200 µatm), the overcalcified morphotypes showed the same growth inhibition (-29.1±6.3%) as moderately calcified morphotypes isolated from non-acidified water (-30.7±8.8%). Under the high CO₂/low pH condition, production rates of particulate organic carbon (POC) increased, while production rates of particulate inorganic carbon (PIC) were maintained or decreased slightly (but not significantly), leading to lowered PIC/POC ratios in all strains. There were no consistent correlations of response intensity with strain origin. The high CO₂/low pH condition affected coccolith morphology equally or more strongly in overcalcified strains compared to moderately calcified strains. High CO₂/low pH conditions appear not to directly select for exceptionally overcalcified morphotypes over other morphotypes directly, but perhaps indirectly by ecologically correlated factors. More generally, these results suggest that oceanic planktonic microorganisms, despite their rapid turn-over and large population sizes, do not necessarily exhibit adaptations to naturally high CO₂ upwellings, and this ubiquitous coccolithophore may be near a limit of its capacity to adapt to ongoing ocean acidification.

## 1 Introduction

Coccolithophores are planktonic single-celled photoautotrophs mostly in the range 3-20 µm and characterized by bearing calcite plates (coccoliths) (Tyrrell and Young, 2009) and represent one of the most abundant and widespread groups of marine eukaryotic phytoplankton (Iglesias-Rodríguez et al., 2002; Litchman et al., 2015). In addition to being important primary producers, coccolithophores contribute most of the calcium carbonate (CaCO₃) precipitation in pelagic systems. Although CaCO₃ precipitation in the surface is a source of CO₂, i.e., the 'carbonate counter pump' (Frankignoulle et al., 1994), CaCO₃ may enhance sinking of organic matter by imposing a ballast effect on sinking aggregates (Armstrong et al., 2002; Sanders et al., 2010). Thus, this plankton functional group has a complex role in ocean carbon cycles. Roughly a third of current anthropogenic CO₂ emissions are being absorbed in the ocean (Sabine et al., 2004), driving a decrease in pH, the

conversion of $CO_3^{2-}$ to $HCO_3^-$, and a drop in saturation states of the $CaCO_3$ minerals aragonite and calcite ($\Omega_{Ar}$, $\Omega_{Ca}$), phenomena collectively termed ocean acidification (OA, Orr et al., 2005). Although most surface waters are expected to remain super-saturated with respect to calcite ($\Omega_{Ca}$>1), which is less soluble than aragonite, the drop in $\Omega_{Ca}$ might still result in decreases in calcite biomineralization (Hofmann and Schellnhuber, 2009). Understanding the response of

coccolithophores to OA is thus needed for predicting how pelagic ecosystems and the relative intensity of the biological carbon pumps will change as atmospheric $CO_2$ continues to increase.

Many studies designed to assess coccolithophores´ responses to low pH have been performed in short-term culture and mesocosm experiments on time-scales of weeks to months, and carbonate systems were usually manipulated to mimic pre-industrial, present and future $CO_2$ levels. Mesocosm studies have shown that North Sea populations of the cosmopolitan and

abundant species *Emiliania huxleyi* are negatively impacted by low pH conditions (Engel et al., 2005; Riebesell et al., 2017). However, a wide range of growth, calcification (PIC) and productivity (POC) responses to high $CO_2$/low pH conditions have been reported in laboratory cultures of *E. huxleyi*, mostly using different regional strains (Riebesell et al., 2000; Iglesias-Rodriguez et al., 2008; Langer et al., 2009; Müller et al., 2015a, 2017; Brady Olson et al., 2017; Jin et al., 2017). According to a recent comprehensive review and meta-analysis (Meyer and Riebesell, 2015), the mean responses of *E. huxleyi* averaged

over 19 studies indicated that high $CO_2$/low pH conditions have a negative effect on PIC quotas and production rates as well as PIC/POC ratios, but no consistent effects on POC quotas and production rates. The response variability among strains of *E. huxleyi* (Langer et al., 2009; Müller et al., 2015a) is also seen within the genus *Calcidiscus* (Diner et al., 2015), and suggests a high potential for genetic adaptation within coccolithophores.

Such adaptive capacity to high $CO_2$/low pH conditions has been suggested for *E. huxleyi* in lab-based long-term

experimental evolution studies (up to 2000 generations) on clonal strains (Lohbeck et al., 2012; Schlüter et al., 2016). It is still difficult to know to which extent such experiments reflect real-world adaptation processes. First, only asexually propagating cells have yet been explored in the lab, while sexual recombination in natural populations is expected to accelerate adaptation (McDonald et al., 2016). Second, calcification is costly and in nature must be maintained by providing benefits to the cell. What these benefits are remains unclear. It has been suggested that coccoliths may provide defense

against grazing or parasites, modify light/UV levels reaching the cell, amongst other proposed functions (Monteiro et al., 2016). The benefits of calcification likely vary among species, and may have changed over the course of evolution or with environmental change. For example, in paleo-oceans, it might have helped alleviate toxicity from Ca2+ when levels reached up to four-fold higher than in the modern ocean during the Cretaceous (Müller et al., 2015b). The long-term and non-linear declines in calcification observed in experimental adaptation to high $CO_2$/low pH (Schlüter et al., 2016) thus might have a

high potential cost if such changes occurred in nature.

Complementary to experimental approaches, observational studies that correlate coccolithophore communities and levels or rates of calcification with variability in carbonate system parameters offer important insights into possible adaptations to high $CO_2$/low pH. Focusing only on *E. huxleyi* and the closely related genus *Gephyrocapsa* (both within the family Noëlaerhabdaceae), a general pattern has been documented of a decreasing calcite mass of coccoliths and coccospheres with

increasing $pCO_2$ for both modern and recent fossil coccolithophores across the world´s ocean basins (Beaufort et al., 2011). This pattern involved shifts away from more heavily calcified *Gephyrocapsa* that dominated assemblages under the lowest $pCO_2$, towards a spectrum of *E. huxleyi* morphotypes that were more abundant under intermediate and high $pCO_2$: *E. huxleyi* 'type A' morphotypes with heavier coccoliths (more calcite per coccolith) dominated *E. huxleyi* populations in waters with intermediate $pCO_2$ while 'type B/C' or 'type C' morphotypes with successively lighter coccoliths, dominated in higher $pCO_2$

waters (Beaufort et al., 2011; Poulton et al., 2011).

Beyond this comparably clear pattern, the survey by Beaufort et al. (2011) also reported one important exception to the general trend: At two sites approaching the Chilean upwelling zone, forms of *E. huxleyi* with exceptionally over-calcified coccoliths dominated in naturally acidified upwelling waters, where $pCO_2$ reaches values more than two-fold higher than the

equilibrium with present-day atmosphereic levels. Similarly, a year-long monthly survey of coccolithophore communities in the Bay of Biscay found that an over-calcified type A form dominated during the winter, when $pCO_2$ was highest, but contributed only a minor part to the *E. huxleyi* populations in summer, when $pCO_2$ was lowest (Smith et al., 2012). One explanation might be that over-calcified morphotypes are especially tolerant to such OA conditions.

The Eastern South Pacific in front of Chile and Peru presents a natural laboratory for investigating such hypotheses regarding organisms´ responses to ocean acidification. The coastal zone is naturally acidified, with surface waters frequently reaching $pCO_2$ levels >1000 μatm and pH values < 7.7 during upwelling events (Friederich et al., 2008; Torres et al., 2011). In this study, we surveyed the coccolithophore communities of the Chilean upwelling zone as well as adjacent coastal and offshore waters with varying $pCO_2$ levels and isolated *E. huxleyi* strains of dominant morphotypes. In lab-based experiments,

three strains showing distinct over-calcification were compared with two moderately calcified type-A morphotypes in terms of their response to altered $CO_2$ and pH  (400 vs. 1200 μatm $pCO_2$) to investigate whether $CO_2$ might indeed be the environmental driver selecting for the extreme overcalcified morphotypes specific to the Chilean coast.

## 2 Materials and Methods

### 2.1 Surveys

An oceanographic cruise (NBP 1305) was conducted on board R/V *Nathaniel B. Palmer* (NBP) during the early austral winter (27 June-22 July 2013) along a transitional zone from coastal to open ocean waters off central-south Peru and north Chile (Fig. 1A). A total of 24 stations were sampled between 22˚ S and 13˚ S and from 70˚ W to 86˚ W (ranging from 47 to 1424 km from the coast). Central Chile coastal surveys were conducted on board the R/V Stella Maris II (Universidad Católica de Norte) during the mid-spring of 2011 (12 October) and 2012 (28 November) and aboard a rented fishing launch

(18-19 November 2012) in the high $pCO_2$ upwelling zone in front of Tongoy Bay (TON), north Chile (~30ºS-72ºW; Fig. 1B). These two coastal surveys consisted of 7 sampling points distributed between 1 and 23 km off the coast. Another coastal sampling was conducted from a small launch (belonging to the Pontificia Universidad Católica de Chile) during the mid-spring of 2012 (10 November), in the upwelling zone in front of El Quisco Bay (QUI ~33ºS-72ºW; Fig. 1B). This coastal survey consisted of 1 sampling point located at 4 km offshore. Finally, one sampling was conducted from a rented

fishing vessel during the mid-spring of 2011 (01 November), in the mesotrophic waters (MES) that surround the Juan Fernández Islands (JF; ~33ºS-78ºW; Fig. 1B).

### 2.2 Physical-chemical oceanographic parameters

During the NBP cruise, temperature and salinity were measured with a SBE 25 CTD (Sea-Bird Scientific, Bellevue, WA, USA) from rosette casts or from the on-board running seawater system equipped with a SBE 45 conductivity sensor and a

SBE 38 temperature sensor (both from Sea-Bird Scientific). During the 2011 cruise on the R/V Stella Maris II, an SBE 19 *plus* CTD was used (data courtesy of B. Yannicelli). In other samplings, an SBE 18 plus CTD was used for water column measurements. On the 29 November 2012 cruise on the R/V Stella Maris II, surface samples were pumped continuously onboard in underway sampling and analysed with a YSI Pro30 salinometer/thermometer (YSI, Yellow Springs, OH, USA). In October 2011 and November 2012, duplicate 500 mL of surface seawater were collect into borosilicate bottles, fixed with

50μL of $HgCl_2$ saturated solution and stored until measurements of total Dissolved Inorganic Carbon (DIC) and Total Alkalinity (TA). TA was determined by potentiometric titration in an open cell (Heraldsson et al., 1997). Standardization was performed and the accuracy was controlled against a certified reference material (CRM Batch 115 bottled on September 2011) supplied by Andrew Dickson (Scripps Institution of Oceanography, http://andrew.ucsd.edu/co2qc/batches.html). The correction factor was approximately 1.002. Precision (variation between replicas) in TA always less than 0.5% (average

0.1%). DIC was determined using a fully automatic dissolved inorganic carbon analyzer (model AS-C3, Apollo SciTech, Newark, DE, USA), with variation between replicates averaging 0.1% (max. 0.3%). All the dissolved carbonate species from

a seawater sample were extracted as $CO_2$ gas by acidification and nitrogen stripping. The $CO_2$ gas was then quantitatively detected with an infra-red LI-7000 $CO_2$ Analyzer (LI-COR Environmental, Lincoln, Nebraska USA). During the expedition off Juan Fernandez (Nov 2011) pH and TA were measured in fixed samples. pH was measured on the Total Ion scale using spectrophotometric detection of m-cresol purple absorption in a 100 mm quartz cell thermally stabilized at 25.0°C (Dickson et al., 2007) with a BioSpec 1600 spectrophotometer (Shimadzu Scientific Instruments, Kyoto, Japan), with pH between replicas varying less than 0.01 units. During the NBP cruise, direct measurements of sea surface $p$CO$_2$ using NDIR detection were obtained from the ship continuous underway data acquisition system (RVDAS; courtesy of Lamont-Doherty Earth Observatory of Columbia University) in addition to TA samples.

Saturation states of aragonite ($\Omega_{Ar}$) as well as calcite ($\Omega_{Ca}$) and other carbonate system parameters were estimated from the DIC-TA pairs (for samplings off the central Chile coast in October 2011 and November 2012), pH-TA (for expedition off Juan Fernandez in November 2011), $p$CO$_2$-TA pairs (for NBP 1305 cruise during June-July 2013) using CO2SYS software (Pierrot et al., 2006) set with Mehrbach solubility constants (Mehrbach et al., 1973) refitted by Dickson and Millero (Dickson and Millero, 1987). Environmental parameters are provided in Table S1.

Mean sea surface temperature and chlorophyll a (Chl a) monthly climatologies (2002-2014) were obtained from the Modis Aqua satellite (Feldman, G. C., C. R. McClain, Ocean Color Web, MODIS Aqua Reprocessing R2014.0, NASA Goddard Space Flight Center. Eds. Kuring, N., Bailey, S. W. 23 Dec. 2015. http://oceancolor.gsfc.nasa.gov/) and plotted using SeaDAS (Baith et al., 2001) version 7.1 for mac OSX.

**2.3 Phytoplankton analyses**

Discrete seawater samples (Niskin bottles) containing planktonic assemblages were collected at various depths within the upper 150 m, depending on depth of the maximum Chl *a* fluorescence (as proxy of phytoplankton) and from the on-board seawater system when Niskin samples were not available. Duplicate 100 mL samples of seawater (previously filtered through 200 µm Nitex mesh) were fixed (final concentration 1% formaldehyde, 0.05% glutaraldehyde, 10 mM borate pH 8.5) and stored at 4ºC until light microscopic examination.

Samples were sedimented in 100 mL Utermöhl chambers for 48 h prior to counting. The absolute abundance of microplankton (20-200 µm in size) and coccolithophores (ranging from 2.5-20 µm in size, but mostly comprised of species within the range 3-10 µm including *E. huxleyi*, several species of the genera *Gephyrocapsa*, and *Calcidiscus leptoporus*) were estimated with an inverted microscope (Olympus CKX41) connected to digital camera (Motic 5.0). For counts of large diatoms, thecate dinoflagellates and others planktonic cells (>50 µm in size), a 20x objective was used. For counts of small diatoms and athecate dinoflagellates (<50 µm in size) a 40x objective was used. For counts of total coccolithophores, a 40x objective was used with cross polarized light (Edmund Optics polarizers 54926 and 53347).

In parallel, duplicate 250 mL samples of seawater were filtered onto polycarbonate filters (0.2 µm pore-size; Millipore), which were dried and stored in Petri dishes until processing for identification of coccolithophore species and *E. huxleyi* morphotypes. A small cut portion of each dried filter was sputter-coated with gold. The identification and relative abundance of coccolithophore species was performed by counting a minimum of 80 coccospheres per sample by scanning electron microscopy using either a TM3000 (Hitachi High-Technologies, Tokyo, Japan) or a Quanta 250 (FEI, Hillsboro, Oregon, USA). Classification followed Young et al. (2003). To estimate the absolute abundances of each species within the Nöelaerhabdaceae family, which are difficult to distinguish by light microscopy, the relative abundance of each Nöelaerhabdaceae species determined by SEM counts was multiplied by the absolute abundance of total Nöelaerhabdaceae cells determined from light microscopy counts. SEM images were also used to measure the min. and max. coccosphere diameters and coccolith lengths of each Nöelaerhabdaceae species (ImageJ software version 1.48 for Mac OSX). Also, *E. huxleyi* cells were categorized according to Young et al. (2003), based on the distal shield and central plate of coccoliths. For analysis, they were grouped further: Lightly calcified coccoliths exhibited delicate distal shield elements that are well

separated from each other extending from the central area to the outer rim, the central element was completely open, and central area elements were either lacking, lath-like, or plate-like (Fig. 2). These corresponded to the morphotypes B, B/C, C and O (Young et al., 2003; Hagino et al., 2011), a grouping that is supported by recent genetic evidence (Krueger-Hadfield et al., 2014). Moderately calcified coccoliths, corresponding to morphotype A (Young et al., 2003; Hagino et al., 2011), showed thicker distal shield elements that were fused near the central area and often at the rim but were otherwise separated, and a grill central area within a cleanly delimited tube. Two over-calcified morphotypes were observed. One corresponded to the morphotype A-overcalcified type reported in the Bay of Biscay (Smith et al., 2012) with coccolith central areas completely covered or nearly completely covered by elements of the central tube, but distal shield elements not fused (here referred to as A_CC). The second, which we refer to as R/overcalcified, corresponded to R morphotype (distal shield elements fused/slits closed), which exhibited a continuous variation from wide and open central area (Young et al., 2003) to the extreme forms, so far reported only in the Eastern South Pacific (Beaufort et al., 2011), where tube elements had completely or partially overgrown the central area.

## 2.4 Isolation of *E. huxleyi* strains

Clonal isolates of coccolithophores were obtained from some stations by isolation of calcified cells using an InFlux Mariner cell sorter as described previously (Von Dassow et al., 2012; Bendif et al., 2016). During the NBP cruise, the InFlux Mariner was in a portable on-board laboratory and isolation of coccolithophores occurred within six hours of sample collection. For other samplings, live seawater samples were hand-carried to Concepción in a cooler with chilled water, and calcified cells were isolated within 24h (without exposure to light or nutrient addition, to minimize possible clonal reproduction between sampling and cell isolation). Calcified strains were identified by SEM and maintained at 15˚ C (Bendif et al., 2016).

## 2.5 Experimental testing of *E. huxleyi* responses to high $CO_2$/low pH

The experiment was performed at the ocean acidification test facility of the Calfuco Marine Laboratory of the Universidad Austral de Chile (Torres et al., 2013). The aim was to investigate the effects of short term exposure to high $CO_2$/low pH conditions similar to those occurring in an upwelling event. The focus was on determining whether there were differences between the heavy calcified morphotypes and moderately calcified morphotypes in response to short-term exposure to $CO_2$, as would be expected to be experienced by phytoplankton cells from surrounding surface waters inoculating recently upwelled water, where both mooring-mounted and drifter-mounted sensors show pulses of high $CO_2$ over periods of about a week (Friederich et al., 2008). Experiments were conducted in temperature-controlled water baths 15˚ C, with light intensities of 75 µmol photons $m^{-2}$ $s^{-1}$ in a 14:10 hour light:dark cycle. Culture media were prepared from seawater collected in wintertime from the Quintay coast (central Chile), aged for >1 month, enriched with 176 µM nitrate, 7.2 µM phosphate, and with trace metals and vitamins as described for K/2 medium (Keller et al., 1987), and sterilized by filtration through 0.2 µm Stericups (Merck-Millipore, Billerica, MA, USA). Strains were acclimated to light and temperature conditions for at least two consecutive culture transfers, maintaining cell density below 200.000 cells $ml^{-1}$ and ensuring exponential growth during the acclimation phase. Prior to inoculation, 4.5 L in 8 L cylindrical clear polycarbonate bottles (Nalgene) were continuously purged with humidified air with a $p$CO_2 of 400 and 1200 µatm for 24-48 hours at the experimental temperature to allow the carbonate system to equilibrate (controlled with pH readings) as described in detail in Torres et al. (2013). When pH values had stabilized, four experimental bottles per strain per treatment were inoculated at an initial density of 800 cells $ml^{-1}$ (day 0), and aeration with the air/$CO_2$ mixes was continued. Daily measures of pH at 25˚ C were made potentiometrically at 25.0°C using a Metrohm 826 pH meter (nominal accuracy +/- 0.003 pH units) (Metrohm, Herisau, Switzerland) with an Aquatrode Plus with Pt1000 (Metrohm 60253100) electrode calibrated with Tris buffer using established methodology (DOE, 1994; Torres et al., 2013). Samples for TA measurement were taken on day 0 and at the end

of the experiment, and measured for calculation of full carbonate chemistry parameters as described above for natural seawater samples.

Daily cell counts were performed from day 2 on using a Neubauer haemocytometer (as cells were too dilute for this method on day 0). Growth rate was calculated as specific growth rate $\mu$ (day$^{-1}$) = $\ln(N_f/N_0)/\Delta t$, where $N_0$ and $N_f$ are the initial and final cell concentrations and $\Delta t$ is the time interval (days). The experimental cultures were harvested before cell concentrations reached 90,000 cells ml$^{-1}$ to minimize changes to the carbonate system from calcification and photosynthesis based on previous studies using R morphotype strains (Rokitta and Rost, 2012). Samples for measurement of particulate organic carbon (POC) and particulate inorganic carbon (PIC) were taken by filtering four 250 ml samples on 47 mm GF/F filters (pre-combusted for overnight at 500˚ C) which were then dried and stored in aluminium envelopes prior to measurement of C content by the Laboratorio de Biogeoquimica y Isotopos Estables Aplicados at the Pontificia Universidad Católica using a Flash EA2000 Elemental Analyzer (Thermo Scientific, Waltham, MA, USA), with a standard error level calculated to be within 0.008 mg C according to linear regression of calibration curves using acetanilide. For each culture, total carbon (TC) was measured on two replicate filters while POC was measured on two replicate filters after treatment by exposure for 4 hours to 12N HCl fumes (Harris et al., 2001; Lorrain et al., 2003). PIC was calculated as the difference between the TC and POC. POC and PIC concentrations were normalized to cell number, and POC and PIC production rates were obtained by multiplying cell normalized POC and PIC quotas with specific growth rates. Samples were filtered and processed as described above for SEM analysis. For flow cytometry, 1.8 ml samples were fixed by adding 0.2 ml of 10% formaldehyde/0.5% glutaraldehyde, 100 mM borate pH 8.5 (which was stored frozen and thawed immediately before use).

### 2.5.1 SEM and flow cytometric assessments and analyses of coccoliths

Morphological analysis was performed on three replicates of each strain with a scanning electron microscope (Quanta 250) and images were analysed by ImageJ. Attached coccoliths were measured following Rosas-Navarro et al. (2016). On average, a total of 606 (min. 418) coccoliths per treatment were analysed. Coccoliths were classified into complete, incomplete and malformed (Rosas-Navarro et al., 2016). In the R/overcalcified strains, fusion of radial elements and the over-growth of inner tube elements of the distal shield complicated finer scale assessments of coccolith formation. Therefore, we were highly conservative in categorizing coccoliths, and grouped incomplete and malformed coccoliths for statistical analysis. Of all coccospheres imaged, only coccoliths were selected for measurement which were oriented upwards (towards the beam) so that coccolith length measurements were not affected by viewing angle. This meant that an average of 68 coccoliths were measured per strain per treatment. Measurements included coccolith length, the total area of the central area (defined by the inner end of distal shield radial elements), and the portion of the central area which was not covered by the inner tube.

Flow cytometry was performed using a BD InFlux equipped with a 488 nm laser and small particle detector with polarization optics. The laser, optics, and stream were aligned using 3 µm Ultra Rainbow Fluorescent Particles (Spherotech, Lake Forest, IL, USA). Trigger was set on forward scatter light with the same polarization as the laser, with trigger level adjusted for each strain to ensure that all detached coccoliths could be detected. Cells were distinguished by red fluorescence (at 692 nm; due to chlorophyll). Detached coccoliths and calcified cells were distinguished as previously described (Von Dassow et al., 2012). Briefly, calcite-containing particles are above the diagonal formed from optically inactive particles on a plot of forward scatter with polarization orthogonal to the laser versus forward scatter with polarization parallel to the laser. Also, calcite containing particles are high in side scatter. Non-calcified cells fall on the diagonal formed by other particles, including cell debris, bacteria (if present), and calibration/alignment particles. Parameters analyzed included the number of detached coccoliths, percentage of calcified cells, relative change in depolarization of forward scatter light by detached coccoliths, and relative changes in red fluorescence (due to chlorophyll) of cells. All samples for a given treatment and strain were run on the same day with the same settings.

**2.6 Statistical analysis**

To test for significant correlations of environmental parameters (including carbonate chemistry) on coccolithophore community composition or *E. huxleyi* morphotype composition in the natural samples, Redundancy Analysis (RDA) was performed (see Supplement). For most analyses, we selected only data from the surface when multiple depths were available (see Supplementary Section S1 for comparison of surface to deeper samples).

Data from experimental results were analysed in Prism 6 (GraphPad Software, Inc., La Jolla, CA, USA) by 2-way ANOVA followed by Sidak post-hoc pairwise analysis with correction for multiple comparison. Prior to testing, PIC/POC ratio was $log_2$-transformed while percentages (e.g. % area, % calcified cells) were expressed as proportions and arcsine-square-root transformed to permit use of parametric testing. Significance was judged at the $p < 0.05$ level.

# 3 Results

## 3.1 Changes in coccolithophore species and *E. huxleyi* morphotypes in natural communities versus oceanographic conditions

Surface pH (< 10 m depth) at sampling sites ranged from 7.73 (in the El Quisco 2012 sampling), to 8.11 (in the JF sampling). In terms of carbonate chemistry, the surface waters of the ESP showed a general pattern of increasing $CO_2$ and decreasing pH as one moves from open ocean waters to the Chilean coastal upwelling zones, however, as expected, waters were never corrosive for calcite (Fig. 3a). More generally, the NBP and JF, as well as TON and QUI surveys were conducted in a relatively low (average 411.2 ± 41.3 µatm; $N_{samples}$= 27) and high (average 696.6 ± 110 µatm; $N_{samples}$= 14) $CO_2$ levels, respectively.

Coccolithophore numerical abundances ranged from $1 \times 10^3$ cells $L^{-1}$ to $76 \times 10^3$ cells $L^{-1}$ (59 total samples) (Fig. 3b). A total of 40 coccolithophore species were found inhabiting the Eastern South Pacific during the sampling period (Table S2). Shannon diversity index ranged from 1.5 down to 0, while Fisher's alpha index ranged from 4.0 down to 0, and both indices showed coccolithophore diversity was lowest in the most acidified natural waters (Fig. 3a-b).

Five species of the Noëlaerhabdaceae family were observed, including *E. huxleyi*, *Gephyrocapsa ericsonii*, *G. muellerae*, *G. oceanica*, *G. parvula*, the last of which was recently re-assigned from the genus *Reticulofenestra* to the genus *Gephyrocapsa* (Bendif et al., 2016). The Noëlaerhabdaceae family numerically dominated all coccolithophore communities observed, representing between 72.2% and 100% (average 94.1% ± 6.9%) of all coccolithophores in all samples observed. The most abundant coccolithophore outside this family was *Calcidiscus leptoporus*, present at 36% of stations and ranging in relative abundance from 0.9% to 25.4% (average 5.6% ± 6.9%). Within the Noëlaerhabdaceae, *E. huxleyi* was found in every sample, and exhibited relative abundances ranging from 15.5% to 100% of total coccolithophores (Fig. 3c). While *E. huxleyi* represented up to 100% of the coccolithophore community in high-$CO_2$ waters on the central Chile coast (stations in groups "TON (2011)", "TON (2012)" and "QUI"), it was observed in lower relative abundances of samples taken both further off shore (NBP samples H01-U2 and JF stations) and to the north (NBP samples BB2a-BB2f), where indices of coccolithophore diversity were generally higher (Fig. 3b-c). *Gephyrocapsa ericsonii* and *G. parvula* were essentially excluded from high-$CO_2$ waters.

R/overcalcified morphotypes dominated *E. huxleyi* populations in high-$CO_2$ waters near the central Chilean coast (samples in groups "TON (2011)", "TON (2012)" and "QUI" in Fig. 3; see also Fig. S3), representing on average 57.2% ± 22.9% (range 11% to 90%) (Fig. 3d). In contrast, moderately calcified A morphotype coccospheres dominated *E. huxleyi* populations in all low-$CO_2$ waters both further off shore (NBP samples H01-U2 and JF stations) and to waters near the coast to the north (NBP stations BB2a-BB2f) (Fig. 3d, Fig. S3). The other overcalcified morphotype A_CC, a form characteristic of the Subtropical Front in the Western Pacific (Cubillos et al., 2007), represented less than 20% of total coccolithophores, and did not follow a

clear pattern. The lightly calcified morphotypes were usually rare except in some of the samples from near Tongoy/Lengua de Vaca Point upwelling (Stations in groups "TON (2011)" and "TON (2012)" in Fig. 3d), where they seemed to be associated with intermediate $CO_2$ levels.

### 3.2 Phenotypes of *E. huxleyi* clonal isolates compared to natural populations from the high $CO_2$ and low $CO_2$ waters

Throughout the field campaigns, a total of 260 Noëlaerhabdaceae isolates were obtained and analyzed morphologically (Table 1; note that strains from stations nearby in time and space have been grouped). There was a bias towards isolating the dominant type within both the Noëlaerhabdaceae and *E. huxleyi* species complex at each station, and only 2% of the maintained isolates were from the *Gephyrocapsa* genus, suggesting that these closely related species are not as readily cultured as *E. huxleyi*. The lightly calcified morphotype also remained poorly represented in culture compared to the natural
communities, and the A_CC type appeared moderately over-represented. However, among the R/overcalcified and moderately calcified A morphotypes, the dominant morphotype obtained in culture always reflected the dominant morphotype in the natural community. Three representative R/overcalcified morphotypes strains, showing different degrees of overlap of the central area, and two representative A morphotype strains from offshore waters were chosen for experimental analysis (Fig. 4).

### 3.3 Responses of different *E. huxleyi* morphotypes to high $CO_2$/low pH

Aeration with $CO_2$/air mixes prior to inoculation successfully equilibrated $pCO_2$ levels, which remained close to target levels throughout the experiment, with final pH values averaging $8.013 \pm 0.029$ under the control condition (400 µatm $pCO_2$) and $7.574 \pm 0.021$ high $CO_2$/low pH condition (1200 µatm $pCO_2$) (Table 2). Seawater remained supersaturated with respect to calcite ($\Omega_{calcite} > 1$) and $\Omega_{calcite}$ values achieved were in a similar range to those seen in the natural waters sampled (Fig. 2),
with final values averaging across strains $\Omega_{calcite} = 3.252 \pm 0.260$ under the control condition and $\Omega_{calcite} = 1.423 \pm 0.077$ for the high $CO_2$/low pH condition (Table 2). Continued aeration and keeping cell concentration below 90.000 cells $ml^{-1}$ was mostly successful in minimizing changes in carbonate system parameter. Averaging the mean values for each strain, alkalinity changed by $-187 \pm 132$ µmol $kg^{-1}$ ($-8.24\% \pm 5.86\%$) in the control condition and $-29 \pm 19$ µmol $kg^{-1}$ ($-1.26\% \pm 0.82\%$) under the high $CO_2$/low pH condition. However, for strain CHC342 the change in alkalinity under the control
condition was larger ($-18.64\% \pm 1.43\%$) (discussed below). This led to a lower final dissolved $CO_2$ (to $12.4 \pm 0.2$ µmol $kg^{-1}$) compared to the other four strains ($15.0 \pm 1.3$ µmol $kg^{-1}$).

High $CO_2$/low pH significantly reduced the growth rate in all strains and there was no significant interaction between strain and high $CO_2$/low pH effects on growth rate (Fig. 5a; see Table 3 for global 2-way ANOVA statistics). High $CO_2$/low pH increased POC quota (POC $cell^{-1}$) in all strains. However, the interaction between strain and high $CO_2$/low pH was
significant (Fig. 5b; Table 3). The increase in POC quota was not significant in moderately calcified strains CHC428 and CHC440, while the hyper-calcified strain CHC342 exhibited the highest POC quota and the highest increase under OA conditions. The effect of high $CO_2$/low pH on the POC production rate varied among strains: High $CO_2$/low pH increased POC production in most strains, except for the moderately calcified strain CHC428 (Fig. 5c; Table 3). However, the change in POC production was significant in post-hoc pairwise comparisons only for CHC342, in which it increased by 116% (p <
0.0001). Although strain CHC342, which exhibited the most overcalcified coccoliths (completely fused distal shield radial elements and central area nearly completely overgrown by tube elements), when all strains were considered neither POC quota nor POC production were consistently different in R/overcalcified vs. A morphotype strains.

PIC/POC ratios dropped under high $CO_2$/low pH in all strains (Fig. 5d). It is notable that the smallest changes in PIC/POC occurred in the two strains of moderately calcified morphotypes originating from offshore, low $pCO_2$ waters, not the strains
with hyper- or heavily calcified morphotypes originating from coastal waters naturally exposed to high $CO_2$/low pH.

However, although the effect of high $CO_2$/low pH condition was globally significant across all strains according to a two-way ANOVA (Table 3), in pairwise post-hoc comparisons the drop in PIC/POC ratio was only significant in CHC360 (p = 0.005). Also, the effect of strain on PIC/POC was not significant and there was no significant interaction between strain and high $CO_2$/low pH (Table 3). PIC quotas varied among strains and the effect of high $CO_2$/low pH also differed among strains (Fig. 5e; Table 3). The highest PIC quota was recorded in the hyper calcified strain CHC342 and the lowest in the moderately calcified strain CHC440. High $CO_2$/low pH increased PIC quota significantly in strain CHC342 (pairwise post-hoc test, p = 0.0039), but did not change PIC quota or the change was not significant in other strains. PIC production varied among strains (Fig. 5f; Table 3) but there were no significant effects of high $CO_2$/low pH or interaction between strain and high $CO_2$/low pH (Table 3).

10  Decreases in alkalinity correlated with PIC (Table 2, Fig. S4). However, for strains CHC342 and CHC440 the drop in alkalinity was more than two fold greater than what would have been predicted from PIC under control conditions (but not under the high $CO_2$/low pH condition) (Supplementary Section S3, Fig. S4). When data from strains CHC342 and CHC440 were excluded, the linear relationship between measured and predicted change in alkalinity was not significantly different than 1:1 (Fig. S4).

15  R/overcalcified coccoliths were not more resistant to high $CO_2$/low pH than A morphotype coccoliths. High $CO_2$/low pH significantly affected at least one morphological parameter measured in all but the A morphotype strain CHC440 (Fig. 4, Fig. 6). The coccosphere diameters did not change significantly under high $CO_2$/low pH in any of the strains (Fig. 6d; Table 4). Coccolith lengths showed inconsistent and mostly insignificant changes among strains. In the global two-way ANOVA comparison, there was an interaction between treatment and strain (Table 4), but the only significant change under high 20  $CO_2$/low pH detected by post-hoc pairwise comparisons between treatments was a small decrease in CHC428 under high $CO_2$/low pH (Fig. 6e; p = 0.0334). The percentage of the central area that was uncovered by inner tube elements increased under OA (Fig. 6f). The significant interaction between strain and treatment (Table 4) indicated that the effect of high $CO_2$/low pH on this parameter varied among strains. It was most pronounced in strains CHC342 and CHC352, where the inner tube elements were heavily over-grown under low $p$CO$_2$, whereas the effect was modest in the moderately calcified 25  strains CHC428 and CHC440 (and not significant in pairwise post-hoc tests of the effect of treatment within these strains; p > 0.05), where the central area was mostly exposed under both conditions. The incidence of incomplete or malformed coccoliths remained very low in all strains and treatments, but high $CO_2$/low pH caused a modest but significant increase (Fig. 6g; Table 4), ranging from between 0 and 1.3% of coccoliths under low $CO_2$ to between 1.4 and 6.6% under high $CO_2$/low pH. This effect was greatest in R/overcalcified morphotype strains CHC342, CHC352, and CHC360, but there was 30  no significant interaction between strain and treatment in the two-way ANOVA when all strains were considered (Table 4). Flow cytometric analysis (see example cytogram in Fig. S5) showed significant changes in several cytometric parameters in response to high $CO_2$/low pH, which in some cases varied among strains (Fig. 7; Table 5). Relative chlorophyll fluorescence was increased significantly in strains CHC360, CHC440, and CHC428, but dropped significantly in CHC352 (Fig. 7a, Table 5). The proportion of cells which were calcified was high (>97%) in all strains under the 400 μatm $CO_2$ control treatment but 35  dropped modestly (0.04 to 7.2%) in all strains in the OA treatment (Fig. 7b). A significant interaction was detected between strain and treatment in the proportion of cells calcified (Table 5), and this drop in response to high $CO_2$/low pH was greatest in strains CHC360 (average change -7.2%) and CHC440 (average change -5.4%), which were the only two strains for which the difference between treatments was judged significant in pairwise post-hoc testing. In the control $CO_2$ treatment, the relative abundance of detached coccoliths, relative to the number of cells, was low (11.9 cell$^{-1}$ to 14.4 cell$^{1}$) in most strains 40  but high (63 ± 34 cell$^{-1}$) in strain CHC440. Despite significant variability among strains in the relative abundance of detached coccoliths, there were no significant changes under high $CO_2$/low pH (Fig. 7c; Table 5). The relative forward scatter depolarization (a proxy for the amount of calcite on a cell, see (Von Dassow et al., 2012) was decreased significantly under high $CO_2$/low pH (Fig. 7d; Table 5), an effect which varied among strains (Table 5) and was strongest in strain

CHC352. The relative scatter depolarization of detached coccoliths was also decreased under high $CO_2$/low pH (Fig. 7e; Table 5), an effect that varied among strains, and was largest in CHC352 and CHC428.

## 4 Discussion

While an increasing number of studies have focused on examining the potential for adaptation to ocean acidification through
long-term laboratory experiments, this study has taken an alternative approach, to test for local adaptation to short-term high $CO_2$/low pH exposure in populations of cosmopolitan phytoplankton found in waters that experience naturally acidified conditions due to upwelling of high $CO_2$ water. A similar approach has recently been taken in a variety of invertebrate species (Padilla-Gamiño et al., 2016; Gaitán-Espitia et al., 2017; Vargas et al., 2017), finding both benthic/mero-planktonic animals, coralline algae, and holoplanktonic copepods do exhibit local adaptation in regions experiencing naturally high
fluctuations in pH and $CO_2$.

This study confirms that R/overcalcified forms of *E. huxleyi* which appear exceptionally robust (as both the central area is extensively overgrown and the distal shield elements are fused) occur in the coastal zone of central to northern Chile. This was previously hinted from two sampling points/times (Beaufort et al., 2011) and now has been documented in separate years. Within the sub-tropical and tropical Eastern South Pacific, the presence of these morphotypes coincides both with
high $CO_2$/low pH (low $\Omega_{calcite}$) as well as with lower temperature (Fig. S3), and it is difficult to separate these two parameters. However, at the lowest end of the *E. huxleyi* temperature range, populations are often found to be dominated by more lightly calcified morphotypes (Cubillos et al., 2007), so a relationship to temperature would have to be very non-linear. More importantly, while a "type A overcalcified" type was reported in winter waters of the Bay of Biscay (Smith et al., 2012) and a "heavily calcified" type "A*" was reported in the Benguela coastal upwelling (Henderiks et al., 2012) (both
exhibiting only overgrowth of the central area by tube elements but not fusion of distal shield elements), the exceptionally robust R/overcalcified forms seen near Chile have not been reported from these other upwelling systems. Therefore, we set out here to test the simplest hypothesis – focusing on a single factor – that these forms may be adapted to resist high $CO_2$/low pH conditions.

The use of targeted flow cytometry and cell sorting was successful in obtaining representatives of the different forms of *E.*
*huxleyi* in mono-culture to test whether the correlation between phenotype and environment indeed reflected local adaptation. Two of the R/overcalcified strains chosen for experimental tests (CHC352 and CHC360) originated from the high $CO_2$ upwelling near Tongoy/Lengua de Vaca Point (Table 1). Strain CHC342 originated from Puñihuil along the outer (western) coast of Chiloe Island (41.9° S). Although we lack carbonate system data from this site, the Chiloe Island is located approximately where the West Wind Drift arrives at the continent and turns north to form the Humboldt Current
System (Thiel et al., 2007), so we considered CHC342, exhibiting a highly overcalcified R morphotype, might represent the southern end of the *E. huxleyi* populations which drift north and experience high $CO_2$/low pH upwelling conditions. We compared these three R/overcalcified strains to two A morphotype strains isolated from low $CO_2$ waters at a site 1000 km from the nearest shore (NBP cruise station H10 in Fig. 1 and Fig. 3). Organisms in such waters are expected to experience very low fluctuations in pH (Hofmann et al., 2011), and so these strains were expected to exhibit low resistance to transient
high $CO_2$/low pH conditions.

The high $CO_2$/low pH condition (1200 μatm $CO_2$) tested was chosen to represent recently upwelled water based on Torres et al. (1999) compared to $CO_2$ levels in non-upwelling surface waters (400 μatm). The high $CO_2$ level of 1200 μatm was also chosen considering previous laboratory studies of the response of *E. huxleyi*: Results of acclimated growth rate in response to short-term changes in the carbonate system manipulated by bubbling have been reported in several studies for two R
morphotype strains isolated from the Tasman Sea (where high $CO_2$ upwelling is not known), of which four studies reported no significant effect on growth rate of intermediate $CO_2$ levels (Langer et al., 2009; Shi et al., 2009; Richier et al., 2011; Rokitta and Rost, 2012) compared to one which reported a decrease at 750 μatm (Iglesias-Rodriguez et al., 2008). For other

*E. huxleyi* strains, results at intermediate $CO_2$ levels are not consistent either among studies or even between strains used in the same study, while all strains tested at higher levels ($\geq$ 950 µatm) have shown slight to pronounced decreases in growth rate (Langer et al., 2009).

The bloom-former *E. huxleyi* is often considered a fast-growing, pioneer phytoplankton species (Paasche, 2002). However, calcification is costly and most evidence suggests it may confer protective or defensive functions (Monteiro et al., 2016). Thus we considered both growth rate and calcification/morphological responses when analyzing potential adaptation. Surprisingly, we found no evidence that the exceptionally robust form was more resistant to high $CO_2$ than moderately calcified forms that seemed to be excluded from the high $CO_2$ upwelling waters.

The high $CO_2$/low pH treatment reduced growth rate in all strains. The decrease in growth rate was accompanied by an increase in POC quota. This might suggest that cells were getting bigger, compensating for a decreased rate of cell division (as the increase in POC production rate was not significant in 4 of the 5 strains tested). However, the decrease in growth rate was also reflected in a decrease in culture *in vivo* fluorescence (data not shown), changes in coccosphere diameter were insignificant, and changes in cellular fluorescence measured by flow cytometry were small and consistent with only a small possible increase in cell biomass (and not in all strains, as CHC352 showed a decrease in this parameter). Among the few previous studies where a period of pre-acclimation to $CO_2$ was not used prior to growth measurements, inconsistent and non-significant effects of growth have been seen in two R morphotype strains NZEH (PLY M219) (Shi et al., 2009) and RCC1216 (Richier et al., 2011). Another study comparing several morphotypes isolated from the Southern Ocean reported that two "A/overcalcified" strains (similar to the R morphotype strain CHC360, but with distal shield radial elements not consistently fused) were relatively resistant to high $CO_2$/low pH treatments compared to both A morphotype and the lighter B/C morphotype in which growth and calcification were strongly inhibited (Müller et al., 2015a). Thus the strains R/overcalcified strains tested here, originating from high $CO_2$ environments, were surprisingly not resistant to high $CO_2$. While caution is warranted in comparing the absolute resistance of the R and R/overcalcified morphotypes tested in this study to those tested in the study by Müller et al. (2015a) even when similar high $CO_2$/low pH treatments were tested, the robust conclusion is that the A morphotypes tested here from the Eastern South Pacific were not more sensitive than the R/overcalcified strains from neighboring high $CO_2$/low pH waters.

In strain CHC342, the POC quota exceeded values previously reported in the literature for the species in response to high $CO_2$/low pH by more than three-fold. This occurred in all four replicates, sampled at the same time as the low $CO_2$ replicates, so we have no evidence for this increase being a technical artefact. The increase in dissolved $CO_2$ in the high $CO_2$/low pH condition compared to the control condition was highest in strain CHC342 due to a higher consumption of alkalinity/DIC. The levels of dissolved $CO_2$ in the control (400 µatm $pCO_2$) condition for all strains fell in a range (12.4-16.6 µmol kg$^{-1}$) that should be saturating for photosynthesis according to one prior study (Buitenhuis et al., 1999), but sub-saturating for POC production according to a more recent study (Bach et al., 2013). The experimental variability noted might have accentuated a variability among strains in the affinity of *E. huxleyi* for photosynthetic carbon uptake. Bach et al (2013) also reported that growth rate was saturated at lower dissolved $CO_2$ levels than POC production. A similar increase in POC quota in response to high $CO_2$ has been reported in *Calcidiscus quadriperforatus* strain RCC 1168 to correlate with the production of transparent exopolysaccharides (TEP) (Diner et al., 2015), and so we suspect that the increase in POC/cell – at least in CHC342 – might correspond partly to increased TEP production under high $CO_2$/low pH.

As expected, PIC quotas varied among strains. CHC342, the strain showing the greatest degree of over-calcification, showed the highest PIC quota. Strain CHC440, the strain showing the coccoliths with the least percentage covering of the central area by the inner tube and the most delicate distal shield rim elements, showed the lowest PIC quota. However, the PIC quotas of CHC352, CHC360, and CHC428 were not different. The numbers of detached coccoliths per cell were similar among those three strains, but coccoliths produced by CHC428 were slightly larger, partly explaining this result. The PIC/POC ratio also did not show consistent differences among morphotypes.

PIC/POC ratios decreased in all strains and all treatments, similar to what has been reported for most of the strains used in most of the previous studies (reviewed in Meyer and Riebesell, 2015). However, in future studies it will be important to understand how TEP production impacts POC and PIC/POC ratios and responds to high $CO_2$/low pH, as has been shown in *Calcidiscus* (Diner et al., 2015). The effect of high $CO_2$/low pH on calcification (PIC quota and PIC production) was

variable among strains, with no clear pattern related to origin or calcification, and none of these modest effects were significant except the increase in PIC quota in CHC342, the most heavily calcified strain. While calcification rate appears to be sensitive to high $CO_2$/low pH in most studies/strains of *E. huxleyi* (Meyer and Riebesell, 2015) despite periods of acclimation, or even adaptation over hundreds of generations (Lohbeck et al., 2012), all of the strains tested here appear to be more similar in this aspect to strains found to exhibit calcification that is relatively resistant to high $CO_2$/low pH (Langer et

al., 2009). We did not see the dramatic loss of calcification (almost all cells were calcified in all strains and treatments) that was reported, for example, in a B/C morphotype from the Southern Ocean in response to high $CO_2$/low pH (Müller et al., 2015a).

We caution that the changes in alkalinity suggested that both strain CHC342 and strain CHC440 may have had approximately two-fold more PIC quota than what was directly measured under the control $CO_2$/pH condition. This could

have occurred if the acidification of total particulate carbon samples did not effectively dissolve all calcium carbonate in these strains. This is surprising, as the geochemical analysis service that performed POC and PIC measurements followed a standard protocol recommended for both plankton samples and carbonate-rich soil samples (Harris et al., 2001; Lorrain et al., 2003) that has been previously used for measuring PIC and POC quotas in *E. huxleyi* (Zondervan et al., 2002; Sciandra et al., 2003). We speculate that perhaps the coccoliths from these strains differ from other strains in organic content in some

way that makes them more resistant to dissolution. For example, recent comparisons have shown that the content and composition of coccolith-associated polysaccharides varies among *E. huxleyi* strains (Lee et al., 2016). However, to our knowledge, we are not aware of such an effect being reported in the literature. In any case, a possible underestimation of PIC quotas in strains CHC342 and CHC440 under the control $CO_2$/pH condition would mean that POC quota was overestimated under that condition, accentuating the increase in POC by high $CO_2$/low pH. Most importantly, it implies these strains may

not have experienced an increase in PIC quota in response to high $CO_2$/low pH, but instead PIC quota either might have actually been maintained or decreased.

The function of coccoliths is still not certain. However, calcification is costly. It is not immediately clear if the proposed role of calcification to alleviate $Ca^{2+}$ toxicity could cause the selection of overcalcified coccoliths in the high $CO_2$/low pH upwelling waters, as the differences in $Ca^{2+}$ concentrations are vanishingly small compared to the levels at which

calcification was observed to show this benefit in the lab (Müller et al., 2015b). Likewise, a possible physiological role of calcification as a carbon-concentrating mechanism to support photosynthesis in low $CO_2$ waters is not supported by the current balance of evidence in published literature for *E. huxleyi* (Trimborn et al., 2007) as reviewed in Monteiro et al. (2016). In any case, such an explanation could not explain why highly calcified cells would be selected for in high $CO_2$ waters. Most evidence suggests calcification may serve protective or defensive functions (Monteiro et al., 2016), in which

case not only the rate of calcification but also the form and quality of the coccoliths would be important. Thus we also considered responses in coccolith morphology when analyzing potential adaptation.

Both microscopic and flow cytometric measures indicated that coccolith morphology was not more resistant to high $CO_2$/low pH conditions in the R/overcalcified strains isolated from naturally high $CO_2$/low pH upwellings than in the A morphotype strains isolated from far offshore waters in equilibrium with the atmosphere, that are not known to experience natural high

$CO_2$/low pH episodes. The increase in the percentages of malformed or incomplete coccoliths in response to high $CO_2$/low pH was most pronounced in the R/overcalcified morphotypes, although these percentages remained low in all strains and both treatments compared to other studies. In other *E. huxleyi* morphotypes, the thickness of the tube around the coccolith central area is reported to decrease modestly under acidification conditions (Bach et al., 2012; Young et al., 2014), and a

similar effect was seen in the two A morphotype strains here. In our study, this effect was most pronounced, resulting in a highly eroded appearance, in the most heavily over-calcified R strains in which the tube overgrows the central area. Coccoliths are formed in intracellular compartments, and the extracellular $\Omega_{Ca}$ remained >1 in the experiments, so this must be due to effects on the formation of coccoliths, not erosion after coccolith secretion. This also shows that the degree of covering of the central area in these types depends on condition in the R morphotype, however, the principal morphotype classification of each of the 5 strains did not change, as expected if morphotype is genetically determined (Young and Westbroek, 1991). The disappearance of the underlying central area ("hollow coccoliths") reported in one study (Lefebvre et al., 2012) was not observed here. The morphological observations by SEM were supported by flow cytometric results, which also showed changes in the relative depolarization of forward scatter light both of whole coccospheres and detached coccoliths.

The observation that the morphology and quality of coccoliths of moderately calcified A morphotype strains were comparatively little affected, while R/overcalcified forms were strongly affected, does not appear consistent with the hypothesis that over-calcification of distal shield elements in the *E. huxleyi* present in naturally acidified high $CO_2$ water serves to compensate for high $CO_2$/low pH effects on coccoliths. Some other factor must select for the R/overcalcified morphotype in the coastal zone of Chile. An A morphotype ("A*") exhibiting partial and irregular extension of inner tube elements over the central area (but not closure of spaces between distal shield radial elements) was dominant in the Benguela upwelling zone (not the more extreme R/overcalcified types) (Henderiks et al., 2012), while the A_CC type, although rare in front of Chile, was dominant in the Northeast Atlantic (Bay of Biscay) in winter (Smith et al., 2012). It is interesting to speculate that high productivity conditions in eastern boundary coasts promote persistent higher abundances of grazers or phytopathogenic bacteria, against which the overcalcified coccoliths might provide better defense.

The lack of evidence for regional scale local adaptation (either in terms of growth or morphology) to short-term high $CO_2$/low pH conditions in *E. huxleyi* populations that are naturally exposed to pulses of naturally high $CO_2$/low pH upwelling conditions, contrasts with the recent findings showing adaptation to ocean acidification in estuarine habitats in invertebrates and coralline algae (Padilla-Gamiño et al., 2016; Gaitán-Espitia et al., 2017; Vargas et al., 2017), including the neritic but holoplanktonic copepod *Acartia tonsa* (Vargas et al., 2017). However, among *E. huxleyi* there is variability in response to high $CO_2$/low pH (Riebesell et al., 2000; Iglesias-Rodriguez et al., 2008; Langer et al., 2009; Meyer and Riebesell, 2015) that appears to correlate with morphotype and origin at least in one study (Müller et al., 2015a). This variability may already have been subject to selection. In that case, perhaps it is appropriate to consider that the present study documented a similar resistance of offshore Eastern South Pacific A morphotype strains to that of coastal strains from high $CO_2$/low pH waters. Although the offshore populations should be exposed less to high $CO_2$/low pH conditions, they might experience such conditions occasionally: Intrathermocline eddies, subsurface lenses between approximately 100 m and 500 m depth, have recently been documented to transport sub-surface waters low in dissolved $O_2$ in an offshore direction in the Eastern South Pacific (Andrade et al., 2014; Combes et al., 2015). These eddies would also be expected to be high $CO_2$/low pH, and intrusions of such water might affect organisms at the base of the euphotic zone. Such processes would still result in lower $CO_2$ exposure than the exposure to high $CO_2$/low pH occurring at the coast. Perhaps the nearshore *E. huxleyi* populations, despite being exposed more frequently and more intensely to high $CO_2$/low pH conditions, have already reached some limit that prevents adaptation to further increases in $CO_2$ which limit the negative effects of these conditions on growth rate, calcification, and coccolithogenesis. Overall, the observation of consistent declines in growth rates, PIC quotas, and PIC/POC ratios, even in genotypes that naturally are exposed to high $CO_2$/low pH conditions, supports the prediction that PIC-associated POC export may decline under future OA conditions, potentially weakening the biological pump (Hofmann and Schellnhuber, 2009).

## Author contributions

PD led the study, carried out sampling in field surveys, performed flow cytometric isolation of *E. huxleyi* strains, carried out statistical analysis of experimental data, supervised processing of samples by flow cytometry and electron microscopy, and wrote the manuscript. FDR conducted characterization of coccolithophore communities and *E. huxleyi* morphotype composition, analysed the relationships of coccolithophore communities and *E. huxleyi* morphotypes to environmental parameters, assisted with part of the high $CO_2$/low pH experiments in the Calfuco Marine Laboratory, and helped prepare the first draft of the manuscript and figures. EMB participated in field studies in 2012, helped with classification of *E. huxleyi* morphotypes, and trained and supervised FDR. in coccolithophore taxonomic classification. JDGE led experimental work in the Calfuco laboratory and provided key comments and editing of the manuscript. SR provided insights into interpretation of results and edited a draft of the manuscript. DM helped plan and perform experiments in the Calfuco lab. UJ assisted with initial plans and later interpretations. RT performed chemical analysis on seawater samples and helped set up the Calfuco Marine Laboratory experiments.

The authors declare they have no conflicts of interest.

## Acknowledgements

This work was supported by the Comisión Nacional de Investigación Científica y Tecnológica of the Chilean Ministry of Education (FONDECYT grants 1110575 and 1141106, and grant CONICYT USA 20120014 to PD, a doctoral fellowship CONICYT-PCHA/Doctorado Nacional/2013- 21130158 to FDR, FONDECYT postdoc grant 312004 to DMF, and FONDEQUIP EQM130267 for the purchase of the InFlux cell sorter), by the Iniciativa Cientifica Milenio of the Chilean Ministry of Economy through the Instituto Milenio de Oceanografía de Chile (grant IC 120019), by the ASSEMBLE program (grant 227799; EMB), and by International Research Network "Diversity, Evolution and Biotechnology of Marine Algae" (GDRI N∘ 0803) of the Centre Nacional de Recherche Scientifique (PD). The authors thank J. Navarro for access to the Calfuco Marine Laboratory, V. Flores for assisting with SEM analysis, J. Beltrán for work as lab manager of the Santiago lab.

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

**Table 1. Noëlaerhabdaceae strains isolated during this study. All sites near Tongoy were grouped in 2011 and in 2012, as were the sites at JF in 2011.**

| Site | Total strains | E. huxleyi | | | | Other species | | |
|---|---|---|---|---|---|---|---|---|
| | | R/over | A_CC | A | Light | G muel. | G eric. | G parv. |
| TON 2011 | 132 | 85% | 10% | 2% | 1% | 2% | 0% | 0% |
| JF 2011 | 34 | 32% | 35% | 32% | 0% | 0% | 0% | 0% |
| TON 2012[a] | 20 | 90% | 10% | 0% | 0% | 0% | 0% | 0% |
| Puñi. 2012[b] | 10 | 40% | 20% | 40% | 0% | 0% | 0% | 0% |
| NBP H1 | 15 | 0% | 33% | 67% | 0% | 0% | 0% | 0% |
| NBP H10[c] | 28 | 0% | 21% | 55% | 24% | 0% | 0% | 0% |
| NBP BB2 | 21 | 0% | 33% | 43% | 0% | 0% | 5% | 19% |

[a] Site represented by strains CHC352 and CHC360

[b] Site represented by strain CHC342

5   [c] Site represented by strains CHC428 and CHC440

**Table 2. Carbonate system parameters during experiment. Means ± standard deviations of experimental replicates at the time of inoculation ($T_{inoc}$ and harvesting ($T_{final}$) are given. pH at the experimental temperature is calculated from measured pH at 25° C. Treatment is specified by $CO_2$ partial pressure (µatm) of air:$CO_2$ mix. $pCO_2$ units are µatm, alkalinity units are µmol kg$^{-1}$. The average ± standard deviations across strains for cell-free control bottles and mean experimental bottle values are also provided. The last two rows give the average and maximum standard deviations between replicates among all strains.**

| Strain | Treat. | $pCO_2$ | | Alkalinity | | pH | | $\Omega_{calcite}$ | | Dissolved [$CO_2$] | |
|---|---|---|---|---|---|---|---|---|---|---|---|
| | | $T_{inoc}$ | $T_{final}$ | $T_{inoc}$ | $T_{final}$ | $T_{inoc}$ | $T_{final}$ | $T_{inoc}$ | $T_{final}$ | $T_{inoc}$ | $T_{final}$ |
| 342 | 400 | 422.0±38 | 332±4 | 2260±7 | 1839±25 | 8.020±0.033 | 8.029±0.033 | 3.531±0.225 | 2.891±0.097 | 15.8±1.4 | **units** |
| | 1200 | 1314±27 | 1257±36 | 2264±5 | 2207±19 | 7.574±0.008 | 7.582±0.008 | 1.402±0.026 | 1.400±0.022 | 50.8±49.4 | **units** |
| 352 | 400 | 402.5±6 | 370.0 * | 2292±13 | 2168 * | 8.042±0.005 | 8.035±0.005 | 3.591±0.041 | 3.494 * | 15.6±0.2 | **units** |
| | 1200 | 1226±27.6 | 1341±65 | 2274±12 | 2161±20 | 7.601±0.007 | 7.561±0.018 | 1.440±0.015 | 1.339±0.057 | 47.6±1.1 | **units** |
| 360 | 400 | 441.4±18.7 | 457.4±47.7 | 2270±6 | 2126±7 | 8.005±0.016 | 7.965±0.040 | 3.552±0.105 | 3.079±0.270 | 16.0±0.7 | **units** |
| | 1200 | 1186±94.7 | 1409±156 | 2289±10 | 2254±4 | 7.623±0.032 | 7.545±0.043 | 1.648±0.107 | 1.370±0.128 | 43.0±3.4 | **units** |
| 428 | 400 | 440.3±21.5 | 418.5±12.9 | 2261±6 | 2157±19 | 8.004±0.018 | 8.004±0.010 | 3.537±0.117 | 3.328±0.035 | 15.9±0.8 | **units** |
| | 1200 | 1259±6.4 | 1247±30.6 | 2262±4 | 2250±5 | 7.592±0.002 | 7.592±0.009 | 1.521±0.007 | 1.494±0.027 | 45.8±0.2 | **units** |
| 440 | 400 | 457.4±26.0 | 381.6±5.6 | 2254±6 | 2114±17 | 7.988±0.021 | 8.033±0.005 | 3.259±0.127 | 3.469±0.104 | 17.3±1.0 | **units** |
| | 1200 | 1487±32.2 | 1249±55.0 | 2261±5 | 2235±7 | 7.522±0.009 | 7.591±0.019 | 1.243±0.022 | 1.512±0.050 | 56.2±1.2 | **units** |
| Ave. w/o cells | 400 | 434.0±37.1 | 396.7±20.7 | 2265±13 | 2273±19 | 8.011±0.032 | 8.051±0.022 | 3.482±0.236 | 3.698±0.124 | 16.2±1.5 | **units** |
| | 1200 | 1286±584.2 | 1290±28.1 | 2268±8 | 2239±58 | 7.585±0.036 | 7.583±0.011 | 1.444±0.135 | 1.461±0.038 | 49.1±4.9 | **units** |
| Ave. with cells | 400 | 432.7±21.0 | 392.0±47.8 | 2267±15 | 2081±137 | 8.012±0.020 | 8.013±0.029 | 3.494±0.133 | 3.252±0.260 | 16.1±0.7 | **units** |
| | 1200 | 1294±117.1 | 1301±72.1 | 2270±12 | 2241±22 | 7.582±0.038 | 7.574±0.021 | 1.451±0.149 | 1.423±0.077 | 48.4±5.0 | **units** |
| Ave. std dev. | 400 | 22.1 | 17.6 | 8 | 17 | 0.018 | 0.016 | 0.123 | 0.127 | 0.8 | **units** |
| | 1200 | 37.5 | 68.6 | 7 | 11 | 0.012 | 0.020 | 0.035 | 0.057 | 1.4 | **units** |
| Max. std dev. | 400 | 38.2 | 47.7 | 13 | 25 | 0.033 | 0.040 | 0.225 | 0.270 | 1.4 | **units** |
| | 1200 | 94.7 | 156 | 12 | 20 | 0.033 | 0.043 | 0.107 | 0.128 | 3.4 | **units** |

*Only one alkalinity sample was analysed from the 400 $p$CO2 treatment for strain CHC352, as three were lost in transit between labs.

**Table 3. Global 2-way ANOVA results for growth and biogeochemical parameters of strains exposed to high $CO_2$/low pH conditions versus control $CO_2$ treatment. PIC/POC values were log2-transformed prior to testing.**

| | | Growth rate | POC | POC prod | PIC | PIC prod | PIC/POC |
|---|---|---|---|---|---|---|---|
| Source of variat. | Interact. | 2.63 % | 21.7 % | 18.8 % | 10.3 % | 6.00 % | 8.15 % |
| | Strain | 13.7 % | 62.8 % | 77.5 % | 71.3 % | 69.2 % | 10.9 % |
| | $CO_2$ | 60.9 % | 23.0 % | 9.67 % | 3.68 % | 2.13 % | 37.8 % |
| F-values | Interact. | $F_{4,29} = 0.926$ | $F_{4,25} = 36.1$ | $F_{4,25} = 27.0$ | $F_{4,25} = 3.08$ | $F_{4,25} = 1.65$ | $F_{4,25} = 1.15$ |
| | Strain | $F_{4,29} = 4.83$ | $F_{4,25} = 105$ | $F_{4,25} = 111.0$ | $F_{4,25} = 21.3$ | $F_{4,25} = 19.0$ | $F_{4,25} = 1.54$ |
| | $CO_2$ | $F_{1,29} = 85.7$ | $F_{1,25} = 153$ | $F_{1,25} = 55.6$ | $F_{1,25} = 4.38$ | $F_{1,25} = 2.33$ | $F_{1,25} = 21.3$ |
| p-values | Interact. | 0.463 | < 0.0001 | < 0.0001 | 0.0343 | 0.194 | 0.358 |
| | Strain | 0.0041 | < 0.0001 | < 0.0001 | < 0.0001 | < 0.0001 | 0.222 |
| | $CO_2$ | < 0.0001 | < 0.0001 | < 0.0001 | 0.0466 | 0.139 | 0.0001 |

5  **Table 4. Global 2-way ANOVA results for coccosphere and coccolith parameters of strains exposed to high $CO_2$/low pH conditions versus control $CO_2$ treatment. Proportions of central area covered and of incomplete or malformed coccoliths were arcsine-squareroot-transformed prior to testing.**

| | | Coccosphere diameter | Coccolith length | Proportion central area covered | Proportion of coccoliths incompl. or malform. |
|---|---|---|---|---|---|
| Source of variat. | Interact. | 7.58 % | 34.7 % | 12.3 % | 4.40 % |
| | Strain | 53.7 % | 25.3 % | 53.3 % | 18.0 % |
| | $CO_2$ | 4.76 % | 0.396 % | 29.2 % | 55.4 % |
| F-values | Interact. | $F_{4,19} = 1.071$ | $F_{4,19} = 4.62$ | $F_{4,19} = 21.9$ | $F_{4,19} = 1.18$ |
| | Strain | $F_{4,19} = 7.595$ | $F_{4,19} = 3.37$ | $F_{4,19} = 94.7$ | $F_{4,19} = 4.83$ |
| | $CO_2$ | $F_{1,19} = 2.689$ | $F_{1,19} = 0.211$ | $F_{1,19} = 207$ | $F_{1,19} = 59.6$ |
| p-values | Interact. | 0.398 | 0.0090 | < 0.0001 | 0.351 |
| | Strain | 0.0008 | 0.0304 | < 0.0001 | 0.0074 |
| | $CO_2$ | 0.118 | 0.652 | < 0.0001 | < 0.0001 |

**Table 5: Global 2-way ANOVA results for flow cytometric parameters. The percentages of calcified cells were expressed as a fraction and arcsine-squareroot-transformed prior to testing.**

| | | Rel. red fluoresce. | % calcified | # detached coccol. | Rel. scatter depol. cells | Rel. scatter depol. detached liths |
|---|---|---|---|---|---|---|
| Source of variat. | Interact. | 35.0 % | 18.9 % | 2.27 % | 11.2 % | 7.85 % |
| | Strain | 34.7 % | 6.42 % | 67.8 % | 55.8 % | 57.9 % |
| | $CO_2$ | 13.5 % | 38.7 % | 1.09 % | 25.4 % | 22.8 % |
| F-values | Interact. | $F_{3,20} = 16.0$ | $F_{3,20} = 3.62$ | $F_{3,20} = 0.525$ | $F_{3,20} = 36.5$ | $F_{3,20} = 12.3$ |
| | Strain | $F_{3,20} = 15.9$ | $F_{3,20} = 1.23$ | $F_{3,20} = 15.7$ | $F_{3,20} = 182$ | $F_{3,20} = 90.6$ |
| | $CO_2$ | $F_{1,20} = 18.5$ | $F_{1,20} = 22.2$ | $F_{1,20} = 0.757$ | $F_{1,20} = 249$ | $F_{1,20} = 107$ |
| p-values | Interact. | < 0.0001 | 0.0309 | 0.670 | < 0.0001 | < 0.0001 |
| | Strain | < 0.0001 | 0.326 | < 0.0001 | < 0.0001 | < 0.0001 |
| | $CO_2$ | 0.0003 | 0.0001 | 0.395 | < 0.0001 | < 0.0001 |

**Figure 1: Map of stations sampled during NBP 1305 cruise (Jun.-Jul. 2013) (a) and in smaller field expeditions of Oct.-Nov. in 2011-2012 (b). SST climatologies (2002-2012) are plotted for the month of July (a) and October (b).**

**Figure 2: The most abundant coccolithophores in the SE Pacific. (a-d) Morphotypes of *E. huxleyi*: Lightly calcified (a), moderately calcified A morphotype (b), morphotype A_CC (c), morphotype R/overcalcified (d). *Gephyrocapsa parvula* (e), *G. ericsonii* (f), *G. muellerae* (g), and *Calcidiscus leptoporus*. Scale bars are 1 μm (a-g) and 3 μm in (h).**

**Figure 3: Environmental conditions, coccolithophore community, and *E. huxleyi* morphotypes. (a) Temperature, pH,**
**$CO_2$, and $\Omega_{calcite}$. (b) Coccolithophore abundance, and Shannon and Fisher's alpha diversity indices. (c) Relative abundance of principal coccolithophore taxa. (d) Relative abundance of *E. huxleyi* morphotypes. The lightly calcified morphotypes B, O, and B/C have been grouped together.**

**Figure 4: Representative coccospheres from each strain and treatment tested in the experiment. CHC342 was isolated**
**from the Pacific coast of Chiloe (41.9° S, 74.0° W) in Nov. 2012, CHC352 and CHC360 were isolated from the Punta Lengua de Vaca upwelling center (30.3° S, 71.7° W) in Nov. 2012. CHC440 and CHC428 were isolated from the farthest west station in the Pacific (station H10, at 16.7° S, 86° W) during the NBP1305 cruise in Jul. 2013.**

**Figure 5: Growth rates (a), POC quotas (b), POC production rates (c), PIC/POC (d), PIC quotas (e), and PIC**
**production rates (f) of *E. huxleyi* strains in response to 400 μatm (black bars) and 1200 μatm (grey bars) $CO_2$ treatments. See Table 3 for global two-way ANOVA results. * indicates significant difference ($p < 0.05$) in pairwise comparison between the two $CO_2$ treatments for a given strain, as judged by Sidak post-hoc testing with correction for multiple comparison.**

**Figure 6: Effects of high $CO_2$/low pH conditions on coccolithophore morphology. (a) Example illustrating coccolith measurements taken including coccolith length (solid line with two arrow heads), total central area/inner tube (TCA) (defined by inner terminal of radial elements), and the part of the central area that is uncovered by tube elements (UCA). (b) Example of a coccolith classified as incomplete/malformed. (c) Example of a very incomplete coccolith (arrow). (d) Coccosphere diameters. (e) Coccolith length. (f) Proportion of central area not covered. (g) Proportion of**
**coccoliths that were malformed or incomplete. See Table 4 for global two-way ANOVA results. * indicates significant difference ($p < 0.05$) in pairwise comparison between the two $CO_2$ treatments for a given strain, as judged by Sidak post-hoc testing with correction for multiple comparison.**

**Figure 7. Effects of high $CO_2$ treatment on flow cytometric properties of cells and detached coccoliths for all**
**treatments and strain. Strain CHC342 is not shown because samples were lost in transit between labs. Shown are the relative fluorescence (compared to control treatment) (a), the proportion of cells that were calcified (b), abundance of detached coccoliths divided by cell abundance (c), relative FSC (scatter depolarization) of cells (d) and detached coccoliths (e). Fluorescence and FSC units are relative and the voltage for the detector for FSC perpendicularly polarized was two-fold higher, resulting in approximately two orders of magnitude higher sensitivity. Scatter**
**depolarization was calculated for every particle as the ratio of FSC with polarizations perpendicular vs parallel to the laser, normalized by the same ratio for non-optically active particles within the same sample. * indicates significant difference ($p < 0.05$) in pairwise comparison between the two $CO_2$ treatments for a given strain, as judged by Sidak post-hoc testing with correction for multiple comparison.**