# Peer review of "Overcalcified forms of the coccolithophore *Emiliania huxleyi* in high CO2 waters are not pre-adapted to ocean acidification."

_Biogeosciences, 2017_

## Referee Comment (RC1) · M. N. Müller (Referee) · 24 Sep 2017

The presented manuscript addresses an important and timely topic within the research area of ocean acidification and the associated impacts on coccolithophore physiology and distribution. The coccolithophore *Emiliania huxleyi* exhibits several different morphotypes which commonly are negatively affected (in terms of growth and calcification rate) under high CO2/low ph conditions. However, a highly calcified strain/ecotype has been repeatedly observed in high CO2/low pH waters like the upwelling region off the coast of Chile. This observation has led to the question if there are certain strains of *E. huxleyi* which present a high resilience against ocean acidification scenarios or even might profit from low pH waters. While strain specific physiological reaction norms have been reported, it is unclear how the highly calcified strain of E. huxleyi, present in low pH waters offshore Chile, responds to ocean acidification scenarios.

The study of von Dassow et al. gives important and crucial answers to these issues. The used methods are appropriate for the goal of this study and the text is well written and transfers the main message to the reader. Therefore, I recommend publication in BG after addressing some minor comments and suggestions.

Comments and suggestions:

1. The use of the term "ocean acidification".
   In the manuscript the differentiation of seawater carbonate chemistry alterations by ocean acidification, upwelling and artificial laboratory manipulation is sometimes not clear and the reader might get the impression that these terms and processes are equivalent to each other.
   The term "ocean acidification" describes the ongoing process of decreasing surface ocean pH induced by anthropogenic carbon dioxide. This global process can be simulated in the laboratory by artificially altering seawater carbonate chemistry. Therefore and strictly speaking, in the laboratory the effects of changing seawater carbonate chemistry are tested. The gained results, however, can be related to ocean acidification but a careful differentiation should be considered.

   For example:
   Abstract on line 28:

"Ocean acidification affected coccolith morphology equally or more strongly in overcalcified strains compared to moderately calcified strains."

As this sentence refers to results from laboratory experiments testing the response to changes in seawater chemistry it should read:

"Low seawater pH conditions affected coccolith morphology equally or more strongly in overcalcified strains compared to moderately calcified strains."

This differentiation might seem pedantic but in my opinion important and the term "ocean acidification" should be solely used for the current globally ongoing anthropogenic process. I understand that the final decision is with the author as it relies on personal preferences and writing style. However, I wanted to raise this point and hope that the authors and readers of this discussion forum will take this point in consideration.

2. At several points in the manuscript (p. 2, line 24 and p. 10, line 2) the authors raises the question on the function/ecological purpose of coccolithophore calcification. This is an important topic and several hypotheses have been put forward.
I agree with the authors that a protective function of coccoliths is not deniable and logic because presumably all organic or inorganic structures located outside the cell membrane will act in some way protective against physicochemical environmental conditions or predation. However, I would like to raise the point that it will be important to differentiate between a possible function/purpose in the modern ocean and the evolutionary trigger of calcification in coccolithophores. The answers to these two questions might be quite different but certainly will help to advance our understanding of coccolithophore calcification. I have addressed this issue previously and invite the interested authors to refer to Muller et al. 2015a.

Müller et al. (2015) Phytoplankton calcification as an effective mechanism to alleviate cellular calcium poisoning, Biogeosciences, 12, 6493-6501.

3. The authors might want to consider moving Fig. 7 to the supplemental material or to the material and method section because it doesn't contribute to the core results and discussion.

4. On a stylistic note I recommend to avoid the start of a sentence with the abbreviation for ocean acidification (e.g. abstract line 28 and 29).

5. Section 2.2.: Please revise the introduction of the abbreviations for dissolved inorganic carbon, total carbon and total alkalinity. The abbreviation should be consistent and introduced on the first appearance.

6. Page 3, line 34: While the accuracy is given for the total alkalinity analysis, it would be helpful if the other could also provide the precision.

7. Page 3, line 35: correct "Bath" to "Batch".

8. Page 6, line 5: see point 5

9. Page 10, line 14-17: The overcalcified strains tested in mentioned study also experienced a reduction of growth rate under elevated pCO2 compared to ambient conditions. The reduction of 5-6% was small but tested significant. Please correct.

10. Page 10, line 32: The words "study" and "strain" are not interchangeable here.

11. Page 10, line 35: Please correct "OA pH".

12. Page 11, line 30 and 36: See point 1.

---

## Referee Comment (RC2) · Anonymous Referee #2 · 7 Nov 2017

The article 'Overcalcified forms of the coccolithophore Emiliania huxleyi in high CO2 waters are not pre-adapted to ocean acidification' by van Dassow et al. is presenting environmental data and its relation to coccolithophorid species composition, in particular Emiliania huxleyi morphology, in the Eastern South Pacific. Furthermore, the theory is tested that over-calcified morphotypes of Emiliania huxleyi, apparently found in high CO2/low pH waters, might be pre-adapted to such conditions in terms of negative effects on cellular calcification rates. The manuscript is well structured and written, and the conclusions backed up by the results. Hence, I support its publications, having only some minor comments and suggestions.

[Figure]

General comments and suggestions:

1) some of the figures, such as Fig. 7 and 8, but also 6, are not essential for the conclusions drawn, thus they could be presented in the supplement.

2) You could consider presenting carbonate chemistry speciation climatologies like the one you have for temperature in Figure 1 (but also see my comment #8 below).

3) Please include carbonate chemistry speciation data from the experiments, potentially in a table (coming to the end of my review I did realise that carbonate chemistry speciation data is indeed included in table 2, but not referenced or discussed). Also, you should report measurement uncertainties for all parameters. Furthermore, did you use certified reference material for the spectrophotometric pH measurements, or how were potential dye impurities quantified?

4) Concerning adjusting pH/pCO2 in the cultures, was the aeration done at the incubation temperature of 15 degrees Celsius? Otherwise, there will be off-sets to target levels.

5) The experimental setup description in the methods section could be more detailed. For instance, it was not clear to me whether there was replication (although I think it is mentioned somewhere in the discussion). Then, it appears that there was a large headspace of about 4 liters in the experimental bottles. That should significantly affect seawater pH/pCO2 over time. Or were the bottles continuously aerated?

6) In Figure 3 there should be no connecting lines for the measurement parameters shown in panel a) and b)

7) The discussion is sometimes difficult to follow when you refer to certain locations, rather than station acronym. So either add these locations to the map or always have a station acronym next to them in the text.

8) What are the implications of your experimental results for observations (and how do they compare to) that coccolith CaCO3 content appears to vary with seawater CaCO3

saturation (e.g. in Beaufort et al. 2011). In this respect, you should also be more precise when you talk about 'levels of calcification' on P2, L32. Along those lines, temperature might be a better candidate (compare Fig. 3a).

5) In the experiments, how do hemocytometer cell counts compare to those done by flowcytometry? And why did you opt to use the former for calculating growth rates rather than the latter?

6) How do PIC based calcification rates and quotas compare to those you could calculate by using the change in alkalinity (potentially corrected for nitrate and phosphate uptake)?

Minor comments and suggestions:

1) P1, L42: The notion that the dissolution of $CaCO_3$ at depth can consume more $CO_2$ than is released at the surface during production is misleading (and I am also not sure what you are aiming at with this statement). It is only the case if you take into account $CO_2$ uptake by photosynthesis at the surface. And if you do that, then you should consider $CO_2$ production through respiration at depth.

2) P3, L30: Replace 'In the 29...' with 'On the 29...'.

3) P5, L25-33: Parts of this could rather go into the discussion.

4) P6, L13: How exactly were POC and TPC measured?

5) P7, L30: What does the notion that 'Emiliania huxleyi abundances correlated with diversity' mean? Also, if that is the case the sentence on P7, L20 should be re-phrased.

6) P8, L11: Looking at Figure 5a, it appears that there was not a statistically significant effect of $pCO_2$ on growth rates in all strains. How does this compare to Table 3?

7) P8, L19: How can POC and PIC quota correlate with strain type in a single strain, i.e. CHC342?

[Figure]

8) P9, L8: How did you distinguish between calcified and naked cells, by flowcytometry?

9) P9, L28: 'exceptionally robust' in which sense?

10) P9, L30: Maybe use the term 'coincides' rather than 'correlates' (see also my general comment #8 above).

11) P10, L 15: How do you explain the differing results found here and in Mueller et al. 2015 regarding the tolerance of over-calcified strains in response to OA?

12) P9, L20: If increased TEP production would be responsible for increased measured cellular POC quotas, one would expect that cell size would not be affected. Do you have evidence for that?

13) P9, L30: Cellular PIC/POC ratios in coccolithophores are not what impacts the biological carbon pump, as most of the POC in the ocean is not produced by diatoms. Re-formulate.

14) P10, L32: Looking at Figure 5d it appears that the decrease in PIC/POC was statistically significant in only one strain. Is that really correct? If so, you should make amendments to the text.

15) P10, L 36: It should read 'quota' not 'quote'.

16) P11, L1: Maybe 'comparison' instead of 'consensus'?

17) P11, L5: do you mean 'percentages'?

18) P11, L8: It should read 'effect'.

19) P11, L15: How could high cell densities and ammonia concentrations explain these differences?

20) P11, L32: 'Resistant' in terms of what?

21) P11, L34: What is your notion, that populations might be already 'near the limit'

based on?

---

## Editor Comment (EC1) · L.J. de Nooijer (Editor) · 22 Nov 2017

Dear Dr Von Dassow and co-authors,

we have now received two reviews on your submitted manuscript. I hope you will use them wisely and formulate a reply and prepare an updated version of your manuscript.

Sincerely,

Lennart de Nooijer

———————————————

---

## Author Comment (AC1) · 5 Dec 2017

Dear Dr. de Nooijer

We are grateful to both reviewers who were supportive of publication of our manuscript "Overcalcified forms of the coccolithophore Emiliania huxleyi in high CO2 waters are not pre-adapted to ocean acidification" in Biogeosciences, but who also made very thorough comments, critically important corrections, and many useful suggested edits. We detail below our point-by-point responses (in the pdf version of this reply, submitted as a supplementary file). We propose to accept almost all of the edits proposed and changes requested by both reviewers, and in all cases we provide the further information, analysis, or clarification requested. We note that we have already prepared detailed information on the carbonate chemistry during experiments in response to Reviewer 2 (we refer to this information in our response to the relevant comment of Reviewer 2). This information will be included with the revised manuscript if you decide that the manuscript should advance.

In the pdf version of this reply, which we submit as a supplementary file, reviewer comments are underlined. Our point-by-point responses are in italics. In some cases, the proposed modification to the text of the manuscript (and occasionally the original text as well) are included to show the modification. In those cases, the text is not italicized, but is placed within quote marks.

Thank you very much for your consideration.

Sincerely yours,

Peter von Dassow and co-authors

Please also note the supplement to this comment:
https://www.biogeosciences-discuss.net/bg-2017-303/bg-2017-303-AC1-supplement.pdf

―――――――――――――――――――――

**Supplement:**

Peter von Dassow
Instituto Milenio de Oceanografía
Facultad de Ciencias Biológicas
Pontificia Universidad Católica de Chile
Avenida Libertador Bernardo O'Higgins #340
Santiago, Chile

Attention: Dr. Lennart de Nooijer, Associate Editor, Biogeosciencies

5 December 2017

Dear Dr. de Nooijer

We are grateful to both reviewers who were supportive of publication of our manuscript "Overcalcified forms of the coccolithophore Emiliania huxleyi in high CO2 waters are not pre-adapted to ocean acidification" in *Biogeosciences*, but who also made very thorough comments, critically important corrections, and many useful suggested edits. We detail below our point-by-point responses (see pdf version of this reply). We propose to accept almost all of the edits proposed and changes requested by both reviewers, and in all cases we provide the further information, analysis, or clarification requested. We note that we have already prepared detailed information on the carbonate chemistry during experiments in response to Reviewer 2 (we refer to this information in our response to the relevant comment of Reviewer 2). This information will be included with the revised manuscript if you decide that the manuscript should advance.

In the pdf version of this reply, which we submit as a supplementary file, reviewer comments are underlined.  Our point-by-point responses are in *italics*.  In some cases, the proposed modification to the text of the manuscript (and occasionally the original text as well) are included to show the modification. In those cases, the text is not italicized, but is placed within quote marks.

Thank you very much for your consideration.

Sincerely yours,

Peter von Dassow and co-authors

**Responses to Reviewer 1:**

… in my opinion important and the term "ocean acidification" should be solely used for the current globally ongoing anthropogenic process. I understand that the final decision is with the author as it relies on personal preferences and writing style. However, I wanted to raise this point and hope that the authors and readers of this discussion forum will take this point in consideration…

*To be most precise, we have changed OA to "high $CO_2$/low pH" throughout the manuscript whenever we refer to this or previous laboratory studies.  The term "high $CO_2$/low pH" is admittedly slightly clumsy, but the vast majority of studies (including this one) were not designed to distinguish between effects of low pH and correlated changes in the carbonate system (e.g., higher $CO_2$, lower carbonate, lowered $\Omega_{calcite}$, alterations in alkalinity or DIC depending on whether the system was manipulated by adding acid or by bubbling with $CO_2$ mixtures). As the discussion of evidence for which of these correlated parameter(s) are important is far beyond the scope of this manuscript, we consider the term "high $CO_2$/low pH" to be most appropriate in its precision.*

2. At several points in the manuscript (p. 2, line 24 and p. 10, line 2) the authors raise the question on the function/ecological purpose of coccolithophore calcification. This is an important topic and several hypotheses have been put forward.

*First sentence in question:*
"The ecological purpose, however, remains unclear, and it has been suggested that the coccosphere may serve for defence against grazing or parasites, for modifying light/UV levels reaching the cell, or even other purposes (Monteiro et al., 2016)."
*Has been modified to:*
"What these benefits are remains unclear. It has been suggested that coccoliths may provide defense against grazing or parasites, modify light/UV levels reaching the cell, amongst other proposed functions (Monteiro et al. 2016). The benefits of calcification likely vary among species, and may have changed over the course of evolution or with environmental change. For example, in paleo-oceans, it might have helped alleviate toxicity from $Ca^{2+}$ when levels reached up to four-fold higher than in the modern ocean during the Cretaceous (Müller et al. 2015)."

*For the second instance cited by Reviewer 1:*
"However, calcification is costly and is most evidence suggests it may serve protective or defensive functions (Monteiro et al., 2016). Thus we considered both growth rate and calcification/morphological responses when analyzing potential adaptation."
*This sentence has been removed from the current paragraph and the issue addressed more completely in a paragraph added just prior to discussion of coccolith morphology results:*
"The function of coccoliths is still not certain. However, calcification is costly. It is not immediately clear if the proposed role of calcification to alleviate $Ca^{2+}$ toxicity could cause the selection of overcalcified coccoliths in the high $CO_2$/low pH upwelling waters, as the differences in $Ca^{2+}$ concentrations are vanishingly small compared to the levels at which calcification was observed to show this benefit in the lab (Müller et al., 2015b). Likewise, a possible physiological

role of calcification as a carbon-concentrating mechanism to support photosynthesis in low $CO_2$ waters is not supported by the current balance of evidence in published literature for *E. huxleyi* (Trimborn et al., 2007) as reviewed in Moneteiro et al. (2016). In any case, such an explanation could not explain why highly calcified cells would be selected for in high $CO_2$ waters. Most evidence suggests calcification may serve protective or defensive functions (Monteiro et al., 2016), in which case not only the rate of calcification but also the form and quality of the coccoliths would be important. Thus we also considered responses in coccolith morphology when analyzing potential adaptation."

3. The authors might want to consider moving Fig. 7 to the supplemental material or to the material and method section because it doesn't contribute to the core results and discussion.
*Both reviewers have requested this change, so it has been carried out, and the figure is now Fig. S4 in supplementary materials.*

4. On a stylistic note I recommend to avoid the start of a sentence with the abbreviation for ocean acidification (e.g. abstract line 28 and 29).
*This was changed when we changed the wording in response to Reviewer 1's comment #1.*

5. Section 2.2.: Please revise the introduction of the abbreviations for dissolved inorganic carbon, total carbon and total alkalinity. The abbreviation should be consistent and introduced on the first appearance.
*We changed "AT" to "TA" throughout.*
*We changed "stored until measurements for Total Carbon (CT) " to "stored until measurements for total Dissolved Inorganic Carbon (DIC)".*
*Change "Total carbon (CT), a.k.a. dissolved inorganic carbon (DIC), was determined…" to " DIC was determined…".*

6. Page 3, line 34: While the accuracy is given for the total alkalinity analysis, it would be helpful if the other could also provide the precision.
*This has now been provided in the revised Methods, not just for alkalinity but we have also been sure to provide this information more clearly for DIC and pH measurements:*
"TA was determined by potentiometric titration in an open cell (Heraldsson et al., 1997). Standardization was performed and the accuracy was controlled against a certified reference material (CRM Batch 115 bottled on September 2011) supplied by Andrew Dickson (Scripps Institution of Oceanography, http://andrew.ucsd.edu/co2qc/batches.html). The correction factor was approximately 1.002. Precision (variation between replicas) in TA always less than 0.5% (average 0.1%). DIC was determined using a fully automatic dissolved inorganic carbon analyzer (model AS-C3, Apollo SciTech, Newark, DE, USA), with variation between replicates averaging 0.1% (max. 0.3%)."
*And*
"pH was measured on the Total Ion scale using spectrophotometric detection of m-cresol purple absorption in a 100 mm quartz cell thermally regulated at 25.0°C (Dickson et al., 2007) with a BioSpec 1600 spectrophotometer (Shimadzu Scientific Instruments, Kyoto, Japan), with

pH between replicas varying less than 0.01 units."

7. Page 3, line 35: correct "Bath" to "Batch".
*This has been corrected.*

8. Page 6, line 5: see point 5
*Done.*

9. Page 10, line 14-17: The overcalcified strains tested in mentioned study also experienced a reduction of growth rate under elevated pCO2 compared to ambient conditions. The reduction of 5-6% was small but tested significant. Please correct.
*We disagree. Two studies are mentioned in those lines. The study by Richier et al. 2011, using RCC1216 states that "growth rate remained unchanged with $\mu = 0.79 \pm 0.02$ and $0.76 \pm 0.02$ for cultures subjected to control and elevated $pCO_2$, respectively" According to the authors, the small drop in growth rate under high $CO_2$ was evaluated to be statistically insignificant. Unfortunately (and this may have caused the confusion), throughout the text of Richier et al. 2011 the use of ">" or "<" appears to have been inverted throughout the text.  In the study of Shi et al. 2009 using PLY M219/NZEH, those authors state "As discussed above, the growth rates of PLY M219 cultures actually increased slightly rather than decreased when we increased pCO2/decreased pH by acid addition, with or without buffer, and remained unchanged upon bubbling of air at pCO2=750µatm (Fig. 2d and Table 1)".  The footnote to their Table 1 reports only significance between treatments (bubbling, acid, or the buffer EPPS) within a pH level, not differences between pH levels. Also, in Shi et al., the changes in growth rate were inconsistent depending on how pH was changed.  It was higher at pH 7.80 than pH 8.10 when acid or EPPS was used, but slightly lower when bubbling was used (with a difference the authors said was insignificant, although statistical results were not completely provided). Thus, our statement that "inconsistent and non-significant effects of growth have been seen…" correctly captures what was observed in these two studies.*

11. Page 10, line 32: The words "study" and "strain" are not interchangeable here.
*Changed to:*
"PIC/POC ratios decreased in all strains and all treatments, similar to what has been reported for most of the strains used in most of the previous studies (reviewed in Meyer and Riebesell 2015)."

12. Page 10, line 35: Please correct "OA pH".
*Changed to "the effect of low pH conditions on calcification…"*

13. Page 11, line 30 and 36: See point 1.
*Change to:*
"The lack of evidence for regional scale local adaptation (either in terms of growth or morphology) to short-term high $CO_2$/low pH conditions in *E. huxleyi* populations that are naturally exposed to pulses of naturally high $CO_2$/low pH upwelling conditions, contrasts with

the recent findings showing adaptation to ocean acidification in estuarine habitats…"

*and*

"Overall, the observation of consistent declines in growth rates, PIC quotas, and PIC/POC ratios, even in genotypes that naturally are exposed to high $CO_2$/low pH conditions, supports the prediction that PIC-associated POC export may decline under future OA conditions, potentially weakening the biological pump (Hofmann and Schellnhuber, 2009)."

**Responses to Reviewer 2:**

1) some of the figures, such as Fig. 7 and 8, but also 6, are not essential for the conclusions drawn, thus they could be presented in the supplement.

*Figure 7 has been moved to the Supplementary Materials as noted above. However, the other two figures present results that are central in the discussion and interpretation, and to final conclusions drawn, and do not belong in supplementary materials.*

2) You could consider presenting carbonate chemistry speciation climatologies like the one you have for temperature in Figure 1 (but also see my comment #8 below).

*Sea surface temperature climatology maps are from satellite data. In contrast, carbonate chemistry speciation can only be measured by direct in situ sampling. While global climatologies have been produced for pCO2, pH, and certain other carbonate parameters, these are produced either by interpolation from the sparse datasets available from cruises and a very limited number of time series stations, or by global ocean/Earth system models (based on such data) and are far too coarse to use at regional scales. Therefore, only the directly measured carbonate chemistry data is presented in Fig. 3. We feel that presenting stations on a map of sea surface temperature climatologies is useful, as it facilitates the reader to get a general sense of how the sampling compares to the major oceanographic gradient of the region.*

3) Please include carbonate chemistry speciation data from the experiments, potentially in a table (coming to the end of my review I did realise that carbonate chemistry speciation data is indeed included in table 2, but not referenced or discussed). Also, you should report measurement uncertainties for all parameters. Furthermore, did you use certified reference material for the spectrophotometric pH measurements, or how were potential dye impurities quantified?

*We have now provided information on accuracy and precision of measurements of alkalinity and pH in the Methods, which were checked against certified reference materials as now clarified. An error was corrected in the methodology for pH measurements in the laboratory experiments: These were performed potentiometrically not spectrophotometrically.*

*Table 2 has been expanded and is now referenced in detail in the first paragraph of the Results section. Discussion of certain points arising from this table are now discussed in the Discussion (see below in response to Comment 6).*

4) Concerning adjusting pH/pCO2 in the cultures, was the aeration done at the incubation temperature of 15 degrees Celsius? Otherwise, there will be off-sets to target levels.

*Yes, the initial aeration was done at the experimental temperature. This is now specified.*

5) The experimental setup description in the methods section could be more detailed. For instance, it was not clear to me whether there was replication (although I think it is mentioned somewhere in the discussion). Then, it appears that there was a large headspace of about 4 liters in the experimental bottles. That should significantly affect seawater pH/pCO2 over time. Or were the bottles continuously aerated?

*These points are now specified:*
"When pH values had stabilized, four experimental bottles per strain per treatment were inoculated at an initial density of 800 cells ml$^{-1}$ (day 0), and aeration with the air/CO$_2$ mixes was continued."

6) In Figure 3 there should be no connecting lines for the measurement parameters shown in panel a) and b)
*This suggested change has been made.*

7) The discussion is sometimes difficult to follow when you refer to certain locations, rather than station acronym. So either add these locations to the map or always have a station acronym next to them in the text.
*In the Results we have followed this suggestion in the two relevant paragraphs:*
"While *E. huxleyi* represented up to 100% of the coccolithophore community in high-CO$_2$ waters on the central Chile coast (stations in groups "TON (2011)", "TON (2012)" and "QUI"), it was observed in lower relative abundances of samples taken both further off shore (NBP samples H01-U2 and JF stations) and to the north (NBP samples BB2a-BB2f),"

"R/overcalcified morphotypes dominated *E. huxleyi* populations in high-CO$_2$ waters near the central Chilean coast (samples in groups "TON (2011)", "TON (2012)" and "QUI" in Fig. 3), representing on average 57.2% ± 22.9% (range 11% to 90%) (Fig. 3d). In contrast, moderately calcified A morphotype coccospheres dominated *E. huxleyi* populations in all low-CO$_2$ waters both further off shore (NBP samples H01-U2 and JF stations) and to waters near the coast to the north (NBP stations BB2a-BB2f). (Fig. 3d)."

*In Discussion:*
"We compared these three R/overcalcified strains to two A morphotype strains isolated from low CO$_2$ waters at a site 1000 km from the nearest shore (NBP cruise station H10 in Fig. 1 and Fig. 3)."

8) What are the implications of your experimental results for observations (and how do they compare to) that coccolith CaCO3 content appears to vary with seawater CaCO3 saturation (e.g. in Beaufort et al. 2011).
Along those lines, temperature might be a better candidate (compare Fig. 3a).
*The reviewer makes an important point here. There is a correlation in our environmental data set between temperature and $\Omega_{calcite}$. In response to these suggestions, we have modified the Discussion with an expanded paragraph about the possible role of temperature vs CaCO3 saturation:*
"This study confirms the presence of exceptionally robust over-calcified forms of *E. huxleyi* in the coastal zone of central to northern Chile. This was previously hinted from two sampling points/times, and now has been documented in separate years (Beaufort et al., 2011). Within the sub-tropical and tropical Eastern South Pacific, the presence of these morphotypes

correlates both with high $CO_2$ (low $\Omega_{calcite}$) as well as with lower temperature, and it is difficult to separate these two parameters. However, at the lowest end of the *E. huxleyi* temperature range, populations are often found to be dominated by more lightly calcified morphotypes (Cubillos et al., 2007), so a relationship to temperature would have to be very non-linear. More importantly, while an "over-calcified A" type was reported in winter waters of the Bay of Biscay (Smith et al., 2012) and a "heavily calcified" type "A*" was reported in the Benguela coastal upwelling (Henderiks et al., 2012), the exceptionally robust forms seen near Chile have not been reported from other upwelling systems. Therefore, we set out here to test the simplest hypothesis – focusing on a single factor – that these forms may be adapted to resist high $CO_2$/low pH conditions."

In this respect, you should also be more precise when you talk about 'levels of calcification' on P2, L32.
*The phrase the line mentioned was changed to:*
"a general pattern has been documented of a decreasing calcite mass of coccoliths and coccospheres with increasing $pCO_2$ was observed"

5) In the experiments, how do hemocytometer cell counts compare to those done by flowcytometry? And why did you opt to use the former for calculating growth rates rather than the latter?
*The flow cytometer could not be transported to the Calfuco experimental facility, so could not be used for daily counts. For that reason, the haemocytometer was used to track growth rates. Unfortunately, the measurements of sample flow rate were not always recorded by the co-author who performed the flow cytometry analysis, and also – as already mentioned – flow cytometry samples were lost for one of the strains tested. Therefore, for clarity, we only use cell abundance data from haemocytometer counts. However, in other work in the lab of author von Dassow, we find an agreement between microscopic and flow cytometric counts (average deviation within ±10%).*
*We did not make a change to the manuscript as this is a minor point that does not impact interpretations of results, and including it may distract the reader by making the text unnecessarily heavy.*

6) How do PIC based calcification rates and quotas compare to those you could calculate by using the change in alkalinity (potentially corrected for nitrate and phosphate uptake)?
*The reviewer has made a very important point and we have now addressed this. At least for strains CHC342 and CHC440, the alkalinity data suggests that PIC may have been substantially underestimated in the high $CO_2$/low pH treatment but not in the control treatment. As the underestimate was relatively consistent across replicas in each case, it seems unlikely that it was an artifact of sampling protocol (e.g., failure to adequately mix the culture prior to sampling). This has now been included in an extensive paragraph in the Results, a brief supplementary text section and a figure in supplementary materials.*
*One possibility is that the coccoliths of these strains under these conditions were resistant to the fuming HCl treatment performed to remove carbonates for measurement of POC. The analytical lab contracted to perform this analysis performed the standard protocol of fuming with 12N HCl*

*for at least 4 hours as recommended both for carbonate rich soil samples and ocean plankton particulate matter samples (Harris et al., 2001; Lorrain et al., 2003). This is a protocol that has been previously used for measuring PIC and POC in coccolithophores even with shorter HCl fuming times (e.g., Zondervan et al., 2002; Sciandra et al., 2003), as cited in new text. We selected this protocol to avoid liberation of organic matter by lysing of cells on the filter. As the total number of cells filtered on each sample was similar to that in other strains tested, we suspect that perhaps the coccoliths in these two strains under the high $CO_2$/low pH treatment might have been somewhat more resistant than the others, perhaps if a higher or different organic content of coccoliths provided some greater protection. We have not seen any report of a similar effect in the literature. However, we note that some other studies (e.g., Langer et al. 2009) follow a protocol that involves adding an HCl solution directly to the filter.*
*We now also include a paragraph in the Discussion where we discuss the implications. If PIC was underestimated in these strains under the control $CO_2$/pH condition, the conclusions we draw would not be altered. In fact, that would accentuate the findings, as these two strains would behave similarly to the other strains.*

Minor comments and suggestions:
1) P1, L42: The notion that the dissolution of CaCO3 at depth can consume more CO2 than is released at the surface during production is misleading (and I am also not sure what you are aiming at with this statement). It is only the case if you take into account CO2 uptake by photosynthesis at the surface. And if you do that, then you should consider CO2 production through respiration at depth.
*This phrase has been removed.*

2) P3, L30: Replace 'In the 29...' with 'On the 29...'.
*Done*

3) P5, L25-33: Parts of this could rather go into the discussion.
*This change has been carried out. Much of this information has been moved to the Discussion in a new paragraph.*

4) P6, L13: How exactly were POC and TPC measured?
*To specify better, the text now reads:*
"Samples for measurement of particulate organic carbon (POC) and particulate inorganic carbon (PIC) were taken by filtering four 250 ml samples on 47 mm GF/F filters (pre-combusted for overnight at 500˚ C) which were then dried and stored in aluminium envelopes prior to measurement of C content by the Laboratorio de Biogeoquimica y Isotopos Estables Aplicados at the Pontificia Universidad Católica using a Flash EA2000 Elemental Analyzer (Thermo Scientific, Waltham, MA, USA), with a standard error level calculated to be within 0.008 mg C according to linear regression of calibration curves using acetanilide. For each culture, total carbon (TC) was measured on two replicate filters while POC was measured on two replicate filters after treatment by exposure for 4 hours to 12N HCl fumes (Harris et al., 2001; Lorrain et al., 2003). PIC was calculated as the difference between the TC and POC."

5) P7, L30: What does the notion that 'Emiliania huxleyi abundances correlated with diversity' mean? Also, if that is the case the sentence on P7, L20 should be re-phrased.
*This has been rephrased to be more clear:*
"While *E. huxleyi* represented up to 100% of the coccolithophore community in high-$CO_2$ waters on the central Chile coast (stations in groups "TON (2011)", "TON (2012)" and "QUI"), it was observed in lower relative abundances of samples taken both further off shore (NBP samples H01-U2 and JF stations) and to the north (NBP samples BB2a-BB2f), where indices of coccolithophore diversity were generally higher (Fig. 3b-c)."

6) P8, L11: Looking at Figure 5a, it appears that there was not a statistically significant effect of pCO2 on growth rates in all strains. How does this compare to Table 3?
*Thanks to the reviewer for catching an error. When preparing the figure, the shared figure legend was placed over Fig. 5a, and covered some of the asterisks marking which strains showed significant effects of treatment on growth rate according to the posthoc test. All strains showed a significant effect.*
7) P8, L19: How can POC and PIC quota correlate with strain type in a single strain, i.e. CHC342?
*Sentence changed to:*
"Although strain CHC342, which exhibited the most overcalcified coccoliths (completely fused distal shield radial elements and central area nearly completely overgrown  by tube elements), when all strains were considered neither POC quota nor POC production were consistently different in R/overcalcified vs. A morphotype strains."

8) P9, L8: How did you distinguish between calcified and naked cells, by flow cytometry?
*The following lines have been added to the Methods to clarify:*
"Briefly, calcite-containing particles are above the diagonal formed from optically inactive particles on a plot of forward scatter with polarization orthogonal to the laser versus forward scatter with polarization parallel to the laser. Also, calcite containing particles are high in side scatter. Non-calcified cells fall on the diagonal formed by other particles, including cell debris, bacteria (if presenent), and calibration/alignment particles."

9) P9, L28: 'exceptionally robust' in which sense?
*This has been now defined:*
"This study confirms that R/overcalcified forms of *E. huxleyi* which appear exceptionally robust (as both the central area is extensively overgrown and the distal shield elements are fused) occur in the coastal zone of central to northern Chile."

10) P9, L30: Maybe use the term 'coincides' rather than 'correlates' (see also my general comment #8 above).
*Recommended change included.*

11) P10, L 15: How do you explain the differing results found here and in Mueller et al. 2015 regarding the tolerance of over-calcified strains in response to OA?
*It is difficult to compare absolute sensitivities among strains in distinct studies, as it has been observed that distinct studies will find different absolute responses even when the same strain is*

*tested. While the observation of Müller et al. (2015b) of no decrease in growth rate in the R morphotype strains tested contrasted with the decrease in growth rate in the R and R/overcalcified morphotypes we tested, Müller et al. did observe a decrease in PIC quota while we did not. Therefore, we cannot conclude with any confidence that the R morphotypes tested by Müller et al. (from the opposite side of the Pacific) were more or less sensitive than those tested here.*

*The more robust comparison is to look at the relative sensitivity of the A morphotype strains vs R morphotype strains in the two studies. Müller et al. (2015a) observed that two A morphotype strains (from the Southern Ocean) were more sensitive than R morphotype strains (from the Western South Pacific). In contrast, we observed that the A morphotype strains from the offshore Eastern South Pacific were equivalently resistant to R/overcalcified morphotypes from high CO2/low pH waters of the coastal Eastern South Pacific.*

*If in fact the different effects of high $CO_2$/low pH on growth rate observed in the two studies are robust, our conclusion that the R/overcalcified morphotypes from high $CO_2$/low pH waters on the Chilean coast have not been able to fully adapt to these conditions is still valid, and means that the local adaptation seen in other types of marine organisms is not occurring for coccolithophores.*

*The lines addressed have been modified as follows to respond to this important comment:*

"Another study comparing several morphotypes isolated from the Southern Ocean reported that two "A/overcalcified" strains (similar to the R morphotype strain CHC360, but with distal shield radial elements not consistently fused) were relatively resistant to high $CO_2$/low pH treatments compared to both A morphotype and the lighter B/C morphotype in which growth and calcification were strongly inhibited (Müller et al., 2015a). Thus the strains R/overcalcified strains tested here, originating from high $CO_2$ environments, were surprisingly not resistant to high $CO_2$. While caution is warranted in comparing the absolute resistance of the R and R/overcalcified morphotypes tested in this study to those tested in the study by Müller et al. (2015a) even when similar high $CO_2$/low pH treatments were tested, the robust conclusion is that the A morphotypes tested here from the Eastern South Pacific were not more sensitive than the R/overcalcified strains from neighboring high $CO_2$/low pH waters."

12) P9, L20: If increased TEP production would be responsible for increased measured cellular POC quotas, one would expect that cell size would not be affected. Do you have evidence for that?

*The only evidence we have is that there was not a significant change in coccosphere diameter (Fig. 6d). However, as it is difficult to measure precisely the thickness of the coccosphere, we cannot rule out that the coccosphere became thinner and the organic cell inside correspondingly increased in diameter (or more dense). For the other strains, we have cytometric data, and if the C:Chl ratio is assumed to remain constant (we have not tested it), then the change (or lack of change) in relative chlorophyll fluorescence could help. Also, most of the forward scatter signal with polarization parallel to the laser is from organic part of the cell (von Dassow et al. 2012) so that would have provided evidence. But unfortunately the samples for strain CHC342 were lost in transit between labs. We do not mention these detailed considerations in the paper as they are beyond the scope extends too long a speculative discussion. We modify the corresponding sentence to:*

"so we suspect that the increase in POC/cell – at least in CHC342 – might correspond partly to increased TEP production under high $CO_2$/low pH"
*We use the words "suspect" and "might" to convey clearly that hard evidence is still lacking.*

13) P9, L30: Cellular PIC/POC ratios in coccolithophores are not what impacts the biological carbon pump, as most of the POC in the ocean is not produced by diatoms. Re-formulate.
*The phrase has been deleted to avoid confusion.*

14) P10, L32: Looking at Figure 5d it appears that the decrease in PIC/POC was statistically significant in only one strain. Is that really correct? If so, you should make amendments to the text.
*It is correct that the decrease in PIC/POC was only significant in one strain (CHC360) in the pairwise posthoc test. However, the global 2-way ANOVA result must be considered prior to the posthoc test. The 2-way ANOVA revealed that – considering all strains – there was a significant effect of treatment on PIC/POC. The ANOVA cannot itself reveal which pairwise differences are significant, only that the trend across all strains was significant. The posthoc test does not rule out that differences in the other strains might be real nor does it reject an overall trend. It merely says that, in the context of multiple testing, those specific differences within the other strains were not significantly different from a difference of 0 by the p-value threshold specified. We have added the words "according to a two-way ANOVA" to remind the reader of the difference between the ANOVA test for whether one or more factors have an effect versus posthoc tests (corrected for multiple comparison) performed after the ANOVA to determine pairwise differences:*
"However, although the effect of high $CO_2$/low pH condition was globally significant across all strains according to a two-way ANOVA (Table 3), in pairwise posthoc comparisons the drop in PIC/POC ratio was only significant in CHC360 (p = 0.005). Also, the effect of strain on PIC/POC was not significant and there was no significant interaction between strain and high $CO_2$/low pH (Table 3)."

15) P10, L 36: It should read 'quota' not 'quote'.
*Changed.*

16) P11, L1: Maybe 'comparison' instead of 'consensus'?
*Simplified to:*
"Both microscopic and flow cytometric measures indicated"

17) P11, L5: do you mean 'percentages'?
*Corrected.*

18) P11, L8: It should read 'effect'.
*Corrected.*

19) P11, L15: How could high cell densities and ammonia concentrations explain these differences?

*We have removed that phrase to avoid distracting from the central point that such an effect was not observed.*

20) P11, L32: 'Resistant' in terms of what?
*We replace the word "resistant" with "response" and with the concept that adaptation may limit "the negative effects of these conditions on growth rate, calcification, and coccolithogenesis"*

21) P11, L34: What is your notion, that populations might be already 'near the limit' based on?
*The onshore populations appear not to be more resistant to high $CO_2$/low pH conditions than the offshore populations, despite being exposed more frequently and more intensely to such conditions. This contradicts the prediction of local adaptation to $CO_2$/pH conditions on a regional scale. However, other lab studies have found that* E. huxleyi *does show high inter-strain variation. In particular, the study of Müller et al. (2015a) found that both growth and calcification were strongly affected by high $CO_2$/low pH conditions in the Southern Ocean B/C morphotype strains. Although we caution that it is difficult to compare absolute sensitivities to high $CO_2$/low pH conditions among strains tested in different studies, the offshore Eastern South Pacific A morphotype strains showed a smaller response in growth rate, calcification, and coccolith morphology than we expected from previous publications studying the response to low pH of other* E. huxleyi *strains. We have conducted extensive modifications to the final paragraph to address this comment and Reviewer 2's comment 10.*

---

## Author Comment (AC3) · 5 Dec 2017

**Overcalcified forms of the coccolithophore *Emiliania huxleyi* in high $CO_2$ waters are not pre-adapted to ocean acidification.**

Peter von Dassow1,2,3\*, Francisco Díaz-Rosas1,2, El Mahdi Bendif4, Juan-Diego Gaitán-Espitia5, Daniella Mella-Flores1, Sebastian Rokitta6, Uwe John6, and Rodrigo Torres7,8

1 Facultad de Ciencias Biológicas, Pontificia Universidad Católica de Chile, Santiago, Chile.

2 Instituto Milenio de Oceanografía de Chile.

3 UMI 3614, Evolutionary Biology and Ecology of Algae, CNRS-UPMC Sorbonne Universités, PUCCh, UACH.

4 Department of Plant Sciences, University of Oxford, OX1 3RB Oxford, UK.

5 CSIRO Oceans and Atmosphere, GPO Box 1538, Hobart 7001, TAS, Australia.

6 Marine Biogeosciences | PhytoChange Alfred Wegener Institute – Helmholtz Centre for Polar and Marine Research, Bremerhaven, Germany.

7Centro de Investigación en Ecosistemas de la Patagonia (CIEP), Coyhaique, Chile.

8 Centro de Investigación: Dinámica de Ecosistemas marinos de Altas Latitudes (IDEAL), Punta Arenas, Chile.

Correspondence to: Peter von Dassow (pvondassow@bio.puc.cl)

**SUPPLEMENTARY MATERIALS**

**Supplementary Section S1**

**Variation in relative abundance of E. huxleyi morphotypes with depth.**

We note that the dominant morphotype of *Emiliania huxleyi* was usually the same at the surface and deeper in the water column (Fig. S1-S2). One exception was a station near Punta Lengua de Vaca (Tongoy Station 18) where lightly calcified morphotypes dominated below the thermocline and R/overcalcified morphotypes dominated above (Fig. S1f). Another exception was the station 2 in the JF survey, where the lightly calcified morphotypes were dominant within and below the picnocline but the A morphotype was dominant, although at the lower total abundance (Fig. S1h). Table S3 (Supplementary section S3) gives abundances with depth at the stations shown in Fig. S1-S2.

---

## Author Response (AR1)

Peter von Dassow
Instituto Milenio de Oceanografía
Facultad de Ciencias Biológicas
Pontificia Universidad Católica de Chile
Avenida Libertador Bernardo O'Higgins #340
Santiago, Chile

Attention: Dr. Lennart de Nooijer, Associate Editor, Biogeosciencies

25 January 2018

Dear Dr. de Nooijer and Biogeosciences Editorial Office;

We are very grateful for the positive final decision on the manuscript and for your helpful final corrections as well. We detail below all the corrections we made, first copying (in green) the corrections identified by the Associate Editor, then listing some other minor corrections or changes made for this final submission.

Thank you very much for your consideration.

Sincerely yours,

Peter von Dassow and co-authors

**Associate Editor Decision: Publish subject to minor revisions (review by editor)** (22 Jan 2018) by Lennart de Nooijer
Comments to the Author:
Dear Dr Von Dassow and co-workers,

I have carefully read your replies to the reviewers and your updated manuscript. Below, I have a few more typographical suggestions, but otherwise your paper is ready for publication in Biogeosciences.

Sincerely,

Lennart

page 2, line 11: sentence seems incomplete
Corrected to "However, a wide range of growth, calcification (PIC) and productivity (POC) responses to high $CO_2$/low pH conditions have been reported in laboratory cultures of *E. huxleyi*, mostly using different regional strains (Riebesell et al., 2000; Iglesias-Rodriguez et al., 2008; Langer et al., 2009; Müller et al., 2015a, 2017; Brady Olson et al., 2017; Jin et al., 2017)."

page 2, line 35: please add e.g. 'for' before 'both'
Corrected

throughout text: italicize the 'p' in 'pCO2'
Corrected. Please note that I had to remove some of the tracking of these and other formatting changes, as my version of Microscoft became very slow tracking the formatting changes.

page 3, line 17: add a space within '22ºS'
Corrected

page 4, line 22: 'um' should be 'μm'
Corrected

page 6, line 4: 'rate' versus 'rates'
Corrected

page 7, line 39: remove the first '.'
Corrected

page 8, line 29: due to copy-pasting 'High CO2/ low pH', the 'High' should be de-capitalized here and there.
Corrected

page 11, line 26: rephrase one of the 'quota'
Corrected sentence to "In strain CHC342, the POC quota exceeded values previously reported in the literature for the species in response to high $CO_2$/low pH by more than three-fold."

page 12, line 2: insert a comma after 'Riebesell'
Corrected

page 12, line 20: please replace 'acid' by 'dissolution'
Corrected

page 13, line 35: replace 'onshore' by 'nearshore'
Corrected

Other corrections:
p. 7 lines 36 and 39 and p. 10 line 3: Referenced supplementary figure S3 as appropriate, e.g.:
"Within the sub-tropical and tropical Eastern South Pacific, the presence of these morphotypes coincides both with high $CO_2$/low pH (low $\Omega_{calcite}$) as well as with lower temperature (Fig. S3)" (p. 10)

p. 10 line 42 to page 11 line 3.
Corrected sentence:

"For other *E. huxleyi* strains, results at intermediate $CO_2$ levels are not consistent either among studies or even between strains used in the same study, while all strains tested at higher levels (≥ 950 µatm) have shown slight to pronounced decreases in growth rate (Langer et al., 2009)." (previously: "For other *E. huxleyi* strains, results at intermediate $CO_2$ levels are not consistent among studies or between strains in the same study, while all strains tested at higher levels (≤ 950 µatm) have shown slight to pronounced decreases in growth rate (Langer et al., 2009).") Corrected

Corrected references (Mendeley does not italicize or use underscore, so I manually italicized species names and converted numbers to underscore as appropriate, e.g., CO2 to $CO_2$).

Corrected Fig. 6f y-axis legend.  Was "proportion covered" and has been corrected to "proportion not covered" (now is consistent with Figure 6 description). Please note that we made some minor adjustments to Fig. 6 in order to combine the multiple panels (including images and graphs) as a single high resolution pdf file, making sure the indications in panel 6a are more easily visible.

All figures have been removed from the .docx file and saved separately inside a compressed .zip folder, following instructions.

[revised manuscript text omitted]

Left:  1 cm, Right:  2.36 cm, Top:  1.65 cm, Bottom:  1.65 cm, Width:  29.7 cm, Height:  20.99 cm, Footer distance from edge:  1.57 cm

| Page 21: [2] Moved from page 21 (Move #1) | Peter von Dassow | 11/29/17 1:03:00 AM |

$p$**CO$_2$ units are µatm, alkalinity units are µmol kg$^{-1}$.**

| Page 21: [3] Formatted | Peter von Dassow | 1/22/18 6:21:00 PM |

Font:Italic

| Page 21: [4] Moved to page 21 (Move #1) | Peter von Dassow | 11/29/17 1:03:00 AM |

 **pCO$_2$ units are µatm, alkalinity units are µmol kg$^{-1}$.**

| Page 21: [5] Deleted | Peter von Dassow | 11/29/17 1:12:00 AM |

| Page 21: [6] Formatted | Peter von Dassow | 1/22/18 6:21:00 PM |

Font:Italic

| Page 21: [7] Formatted | Peter von Dassow | 11/29/17 10:08:00 AM |

Font:10 pt, Not Superscript/ Subscript

| Page 21: [8] Formatted | Peter von Dassow | 11/29/17 10:37:00 AM |

Indent: Left:  -0.13 cm, Right:  0.02 cm

| Page 21: [9] Formatted | Peter von Dassow | 11/29/17 12:36:00 AM |

Indent: Left:  -0.34 cm, Right:  -0.3 cm

| Page 21: [10] Formatted | Peter von Dassow | 11/28/17 10:24:00 PM |

Indent: Left:  -0.14 cm, Right:  -0.14 cm

| Page 21: [11] Formatted | Peter von Dassow | 11/29/17 12:49:00 AM |

Indent: Left:  -0.19 cm, Right:  -0.24 cm

| Page 21: [12] Formatted | Peter von Dassow | 11/29/17 10:08:00 AM |

Font:10 pt

| Page 21: [13] Formatted | Peter von Dassow | 11/29/17 10:08:00 AM |

Font:10 pt

| Page 21: [14] Formatted | Peter von Dassow | 11/29/17 10:37:00 AM |

Indent: Left:  -0.35 cm, Right:  0.02 cm

| Page 21: [15] Formatted | Peter von Dassow | 11/29/17 10:37:00 AM |

Indent: Left:  -0.34 cm, Right:  -0.01 cm

| Page 21: [16] Formatted | Peter von Dassow | 11/29/17 10:38:00 AM |

Centered, Indent: Left:  -0.14 cm

| Page 21: [17] Formatted | Peter von Dassow | 11/29/17 10:38:00 AM |
|---|---|---|

Indent: Left:  -0.11 cm

| Page 21: [18] Formatted | Peter von Dassow | 11/29/17 10:38:00 AM |
|---|---|---|

Indent: Left:  -0.19 cm

| Page 21: [19] Formatted | Peter von Dassow | 11/29/17 10:39:00 AM |
|---|---|---|

Indent: Left:  -0.19 cm

| Page 21: [20] Formatted | Peter von Dassow | 11/29/17 10:39:00 AM |
|---|---|---|

Indent: Left:  -0.17 cm

| Page 21: [21] Formatted | Peter von Dassow | 11/29/17 10:39:00 AM |
|---|---|---|

Indent: Left:  -0.19 cm

| Page 21: [22] Formatted | Peter von Dassow | 11/29/17 10:40:00 AM |
|---|---|---|

Right, Right:  0 cm

| Page 21: [23] Formatted | Peter von Dassow | 11/29/17 10:08:00 AM |
|---|---|---|

Font:10 pt

| Page 21: [24] Formatted | Peter von Dassow | 11/29/17 10:08:00 AM |
|---|---|---|

Font:10 pt

| Page 21: [25] Formatted | Peter von Dassow | 11/29/17 10:37:00 AM |
|---|---|---|

Indent: Left:  -0.35 cm, Right:  0.02 cm

| Page 21: [26] Formatted | Peter von Dassow | 11/29/17 10:37:00 AM |
|---|---|---|

Indent: Left:  -0.34 cm, Right:  -0.01 cm

| Page 21: [27] Formatted | Peter von Dassow | 11/29/17 10:38:00 AM |
|---|---|---|

Centered, Indent: Left:  -0.14 cm

| Page 21: [28] Formatted | Peter von Dassow | 11/29/17 10:38:00 AM |
|---|---|---|

Indent: Left:  -0.11 cm

| Page 21: [29] Formatted | Peter von Dassow | 11/29/17 10:38:00 AM |
|---|---|---|

Indent: Left:  -0.19 cm

| Page 21: [30] Formatted | Peter von Dassow | 11/29/17 10:39:00 AM |
|---|---|---|

Indent: Left:  -0.19 cm

| Page 21: [31] Formatted | Peter von Dassow | 11/29/17 10:39:00 AM |
|---|---|---|

Indent: Left:  -0.17 cm

| Page 21: [32] Formatted | Peter von Dassow | 11/29/17 10:39:00 AM |
|---|---|---|

Indent: Left:  -0.19 cm

| Page 21: [33] Formatted | Peter von Dassow | 11/29/17 10:40:00 AM |
|---|---|---|

Right, Right:  0 cm

| Page 21: [34] Formatted | Peter von Dassow | 11/29/17 10:08:00 AM |
|---|---|---|

Font:10 pt

| Page 21: [35] Formatted | Peter von Dassow | 11/29/17 10:08:00 AM |
|---|---|---|

Font:10 pt

| Page 21: [36] Formatted | Peter von Dassow | 11/29/17 10:37:00 AM |
|---|---|---|

Indent: Left:  -0.35 cm, Right:  0.02 cm

| Page 21: [37] Formatted | Peter von Dassow | 11/29/17 10:37:00 AM |
|---|---|---|

Indent: Left:  -0.34 cm, Right:  -0.01 cm

| Page 21: [38] Formatted | Peter von Dassow | 11/29/17 10:38:00 AM |
|---|---|---|

Centered, Indent: Left:  -0.14 cm

| Page 21: [39] Formatted | Peter von Dassow | 11/29/17 10:38:00 AM |
|---|---|---|

Indent: Left:  -0.11 cm

| Page 21: [40] Formatted | Peter von Dassow | 11/29/17 10:38:00 AM |
|---|---|---|

Indent: Left:  -0.19 cm

| Page 21: [41] Formatted | Peter von Dassow | 11/29/17 10:39:00 AM |
|---|---|---|

Indent: Left:  -0.19 cm

| Page 21: [42] Formatted | Peter von Dassow | 11/29/17 10:39:00 AM |
|---|---|---|

Indent: Left:  -0.17 cm

| Page 21: [43] Formatted | Peter von Dassow | 11/29/17 10:39:00 AM |
|---|---|---|

Indent: Left:  -0.19 cm

| Page 21: [44] Formatted | Peter von Dassow | 11/29/17 10:40:00 AM |
|---|---|---|

Right, Right:  0 cm

| Page 21: [45] Formatted | Peter von Dassow | 11/29/17 10:08:00 AM |
|---|---|---|

Font:10 pt

| Page 21: [46] Formatted | Peter von Dassow | 11/29/17 10:08:00 AM |
|---|---|---|

Font:10 pt

| Page 21: [47] Formatted | Peter von Dassow | 11/29/17 10:37:00 AM |
|---|---|---|

Indent: Left:  -0.35 cm, Right:  0.02 cm

| Page 21: [48] Formatted | Peter von Dassow | 11/29/17 10:37:00 AM |
|---|---|---|

Indent: Left:  -0.34 cm, Right:  -0.01 cm

| Page 21: [49] Formatted | Peter von Dassow | 11/29/17 10:38:00 AM |
|---|---|---|

Centered, Indent: Left: -0.14 cm

| Page 21: [50] Formatted | Peter von Dassow | 11/29/17 10:38:00 AM |
|---|---|---|

Indent: Left: -0.11 cm

| Page 21: [51] Formatted | Peter von Dassow | 11/29/17 10:38:00 AM |
|---|---|---|

Indent: Left: -0.19 cm

| Page 21: [52] Formatted | Peter von Dassow | 11/29/17 10:39:00 AM |
|---|---|---|

Indent: Left: -0.19 cm

| Page 21: [53] Formatted | Peter von Dassow | 11/29/17 10:39:00 AM |
|---|---|---|

Indent: Left: -0.17 cm

| Page 21: [54] Formatted | Peter von Dassow | 11/29/17 10:39:00 AM |
|---|---|---|

Indent: Left: -0.19 cm

| Page 21: [55] Formatted | Peter von Dassow | 11/29/17 10:40:00 AM |
|---|---|---|

Right, Right: 0 cm

| Page 21: [56] Formatted | Peter von Dassow | 11/29/17 10:08:00 AM |
|---|---|---|

Font:10 pt

| Page 21: [57] Formatted | Peter von Dassow | 11/29/17 10:08:00 AM |
|---|---|---|

Font:10 pt

| Page 21: [58] Formatted | Peter von Dassow | 11/29/17 10:37:00 AM |
|---|---|---|

Indent: Left: -0.35 cm, Right: 0.02 cm

| Page 21: [59] Formatted | Peter von Dassow | 11/29/17 10:37:00 AM |
|---|---|---|

Indent: Left: -0.34 cm, Right: -0.01 cm

| Page 21: [60] Formatted | Peter von Dassow | 11/29/17 10:38:00 AM |
|---|---|---|

Centered, Indent: Left: -0.14 cm

| Page 21: [61] Formatted | Peter von Dassow | 11/29/17 10:38:00 AM |
|---|---|---|

Indent: Left: -0.11 cm

| Page 21: [62] Formatted | Peter von Dassow | 11/29/17 10:38:00 AM |
|---|---|---|

Indent: Left: -0.19 cm

| Page 21: [63] Formatted | Peter von Dassow | 11/29/17 10:39:00 AM |
|---|---|---|

Indent: Left: -0.19 cm

| Page 21: [64] Formatted | Peter von Dassow | 11/29/17 10:39:00 AM |
|---|---|---|

Indent: Left: -0.17 cm

**Page 21: [65] Formatted**      **Peter von Dassow**      **11/29/17 10:39:00 AM**

Indent: Left: -0.19 cm

**Page 21: [66] Formatted**      **Peter von Dassow**      **11/29/17 10:40:00 AM**

Right, Right: 0 cm

**Page 21: [67] Formatted**      **Peter von Dassow**      **11/29/17 10:08:00 AM**

Font:10 pt

**Page 21: [68] Formatted**      **Peter von Dassow**      **11/29/17 10:08:00 AM**

Font:10 pt

**Page 21: [69] Formatted**      **Peter von Dassow**      **11/29/17 10:37:00 AM**

Indent: Left: -0.35 cm, Right: 0.02 cm

**Page 21: [70] Formatted**      **Peter von Dassow**      **11/29/17 10:37:00 AM**

Indent: Left: -0.34 cm, Right: -0.01 cm

**Page 21: [71] Formatted**      **Peter von Dassow**      **11/29/17 10:38:00 AM**

Centered, Indent: Left: -0.14 cm

**Page 21: [72] Formatted**      **Peter von Dassow**      **11/29/17 10:38:00 AM**

Indent: Left: -0.11 cm

**Page 21: [73] Formatted**      **Peter von Dassow**      **11/29/17 10:38:00 AM**

Indent: Left: -0.19 cm

**Page 21: [74] Formatted**      **Peter von Dassow**      **11/29/17 10:39:00 AM**

Indent: Left: -0.19 cm

**Page 21: [75] Formatted**      **Peter von Dassow**      **11/29/17 10:39:00 AM**

Indent: Left: -0.17 cm

**Page 21: [76] Formatted**      **Peter von Dassow**      **11/29/17 10:39:00 AM**

Indent: Left: -0.19 cm

**Page 21: [77] Formatted**      **Peter von Dassow**      **11/29/17 10:40:00 AM**

Right, Right: 0 cm

**Page 21: [78] Formatted**      **Peter von Dassow**      **11/29/17 10:08:00 AM**

Font:10 pt

**Page 21: [79] Formatted**      **Peter von Dassow**      **11/29/17 10:08:00 AM**

Font:10 pt

**Page 21: [80] Formatted**      **Peter von Dassow**      **11/29/17 10:37:00 AM**

Indent: Left: -0.35 cm, Right: 0.02 cm

| Page 21: [81] Formatted | Peter von Dassow | 11/29/17 10:37:00 AM |
|---|---|---|

Indent: Left:  -0.34 cm, Right:  -0.01 cm

| Page 21: [82] Formatted | Peter von Dassow | 11/29/17 10:38:00 AM |
|---|---|---|

Centered, Indent: Left:  -0.14 cm

| Page 21: [83] Formatted | Peter von Dassow | 11/29/17 10:38:00 AM |
|---|---|---|

Indent: Left:  -0.11 cm

| Page 21: [84] Formatted | Peter von Dassow | 11/29/17 10:38:00 AM |
|---|---|---|

Indent: Left:  -0.19 cm

| Page 21: [85] Formatted | Peter von Dassow | 11/29/17 10:39:00 AM |
|---|---|---|

Indent: Left:  -0.19 cm

| Page 21: [86] Formatted | Peter von Dassow | 11/29/17 10:39:00 AM |
|---|---|---|

Indent: Left:  -0.17 cm

| Page 21: [87] Formatted | Peter von Dassow | 11/29/17 10:39:00 AM |
|---|---|---|

Indent: Left:  -0.19 cm

| Page 21: [88] Formatted | Peter von Dassow | 11/29/17 10:40:00 AM |
|---|---|---|

Right, Right:  0 cm

| Page 21: [89] Formatted | Peter von Dassow | 11/29/17 10:08:00 AM |
|---|---|---|

Font:10 pt

| Page 21: [90] Formatted | Peter von Dassow | 11/29/17 10:08:00 AM |
|---|---|---|

Font:10 pt

| Page 21: [91] Formatted | Peter von Dassow | 11/29/17 10:37:00 AM |
|---|---|---|

Indent: Left:  -0.35 cm, Right:  0.02 cm

| Page 21: [92] Formatted | Peter von Dassow | 11/29/17 10:37:00 AM |
|---|---|---|

Indent: Left:  -0.34 cm, Right:  -0.01 cm

| Page 21: [93] Formatted | Peter von Dassow | 11/29/17 10:38:00 AM |
|---|---|---|

Centered, Indent: Left:  -0.14 cm

| Page 21: [94] Formatted | Peter von Dassow | 11/29/17 10:38:00 AM |
|---|---|---|

Indent: Left:  -0.11 cm

| Page 21: [95] Formatted | Peter von Dassow | 11/29/17 10:38:00 AM |
|---|---|---|

Indent: Left:  -0.19 cm

| Page 21: [96] Formatted | Peter von Dassow | 11/29/17 10:39:00 AM |
|---|---|---|

Indent: Left:  -0.19 cm

**Page 21: [97] Formatted**      **Peter von Dassow**      **11/29/17 10:39:00 AM**

Indent: Left: -0.17 cm

**Page 21: [98] Formatted**      **Peter von Dassow**      **11/29/17 10:39:00 AM**

Indent: Left: -0.19 cm

**Page 21: [99] Formatted**      **Peter von Dassow**      **11/29/17 10:40:00 AM**

Right, Right: 0 cm

**Page 21: [100] Formatted**      **Peter von Dassow**      **11/29/17 10:08:00 AM**

Font:10 pt

**Page 21: [101] Formatted**      **Peter von Dassow**      **11/29/17 10:08:00 AM**

Font:10 pt

**Page 21: [102] Formatted**      **Peter von Dassow**      **11/29/17 10:37:00 AM**

Indent: Left: -0.35 cm, Right: 0.02 cm

**Page 21: [103] Formatted**      **Peter von Dassow**      **11/29/17 10:37:00 AM**

Indent: Left: -0.34 cm, Right: -0.01 cm

**Page 21: [104] Formatted**      **Peter von Dassow**      **11/29/17 10:38:00 AM**

Centered, Indent: Left: -0.14 cm

**Page 21: [105] Formatted**      **Peter von Dassow**      **11/29/17 10:38:00 AM**

Indent: Left: -0.11 cm

**Page 21: [106] Formatted**      **Peter von Dassow**      **11/29/17 10:38:00 AM**

Indent: Left: -0.19 cm

**Page 21: [107] Formatted**      **Peter von Dassow**      **11/29/17 10:39:00 AM**

Indent: Left: -0.19 cm

**Page 21: [108] Formatted**      **Peter von Dassow**      **11/29/17 10:39:00 AM**

Indent: Left: -0.17 cm

**Page 21: [109] Formatted**      **Peter von Dassow**      **11/29/17 10:39:00 AM**

Indent: Left: -0.19 cm

**Page 21: [110] Formatted**      **Peter von Dassow**      **11/29/17 10:40:00 AM**

Right, Right: 0 cm

**Page 21: [111] Formatted**      **Peter von Dassow**      **11/29/17 10:08:00 AM**

Font:10 pt

**Page 21: [112] Formatted**      **Peter von Dassow**      **11/29/17 10:08:00 AM**

Font:10 pt

**Page 21: [113] Formatted**       **Peter von Dassow**       **11/29/17 10:37:00 AM**

Indent: Left:  -0.35 cm, Right:  0.02 cm

**Page 21: [114] Formatted**       **Peter von Dassow**       **11/29/17 10:37:00 AM**

Indent: Left:  -0.34 cm, Right:  -0.01 cm

**Page 21: [115] Formatted**       **Peter von Dassow**       **11/29/17 10:38:00 AM**

Centered, Indent: Left:  -0.14 cm

**Page 21: [116] Formatted**       **Peter von Dassow**       **11/29/17 10:38:00 AM**

Indent: Left:  -0.11 cm

**Page 21: [117] Formatted**       **Peter von Dassow**       **11/29/17 10:38:00 AM**

Indent: Left:  -0.19 cm

**Page 21: [118] Formatted**       **Peter von Dassow**       **11/29/17 10:39:00 AM**

Indent: Left:  -0.19 cm

**Page 21: [119] Formatted**       **Peter von Dassow**       **11/29/17 10:39:00 AM**

Indent: Left:  -0.17 cm

**Page 21: [120] Formatted**       **Peter von Dassow**       **11/29/17 10:39:00 AM**

Indent: Left:  -0.19 cm

**Page 21: [121] Formatted**       **Peter von Dassow**       **11/29/17 10:40:00 AM**

Right, Right:  0 cm

**Page 21: [122] Formatted**       **Peter von Dassow**       **11/29/17 10:08:00 AM**

Font:10 pt

**Page 21: [123] Formatted**       **Peter von Dassow**       **11/29/17 10:08:00 AM**

Font:10 pt

**Page 21: [124] Formatted**       **Peter von Dassow**       **11/29/17 8:40:00 AM**

Font:10 pt, Font color: Black

**Page 21: [125] Formatted**       **Peter von Dassow**       **11/29/17 10:37:00 AM**

Right, Right:  0.02 cm

**Page 21: [126] Formatted**       **Peter von Dassow**       **11/29/17 10:37:00 AM**

Right, Right:  -0.01 cm

**Page 21: [127] Formatted**       **Peter von Dassow**       **11/29/17 10:38:00 AM**

Right:  0 cm

**Page 21: [128] Formatted**       **Peter von Dassow**       **11/29/17 10:38:00 AM**

Right, Right:  0 cm

| Page 21: [129] Formatted | Peter von Dassow | 11/29/17 10:08:00 AM |
|---|---|---|

Font:10 pt

| Page 21: [130] Formatted | Peter von Dassow | 11/29/17 10:08:00 AM |
|---|---|---|

Font:10 pt

| Page 21: [131] Formatted | Peter von Dassow | 11/29/17 10:37:00 AM |
|---|---|---|

Right, Right:  0.02 cm

| Page 21: [132] Formatted | Peter von Dassow | 11/29/17 10:37:00 AM |
|---|---|---|

Right, Right:  -0.01 cm

| Page 21: [133] Formatted | Peter von Dassow | 11/29/17 10:38:00 AM |
|---|---|---|

Right:  0 cm

| Page 21: [134] Formatted | Peter von Dassow | 11/29/17 10:38:00 AM |
|---|---|---|

Right, Right:  0 cm

| Page 21: [135] Formatted | Peter von Dassow | 11/29/17 10:08:00 AM |
|---|---|---|

Font:10 pt

| Page 21: [136] Formatted | Peter von Dassow | 11/29/17 10:08:00 AM |
|---|---|---|

Font:10 pt

| Page 21: [137] Formatted | Peter von Dassow | 11/29/17 8:48:00 AM |
|---|---|---|

Font:10 pt

| Page 21: [138] Formatted | Peter von Dassow | 11/29/17 10:37:00 AM |
|---|---|---|

Right, Right:  0.02 cm

| Page 21: [139] Formatted | Peter von Dassow | 11/29/17 8:45:00 AM |
|---|---|---|

Font:10 pt

| Page 21: [140] Formatted | Peter von Dassow | 11/29/17 10:37:00 AM |
|---|---|---|

Right, Right:  -0.01 cm

| Page 21: [141] Formatted | Peter von Dassow | 11/29/17 8:45:00 AM |
|---|---|---|

Font:10 pt

| Page 21: [142] Formatted | Peter von Dassow | 11/29/17 10:38:00 AM |
|---|---|---|

Right:  0 cm

| Page 21: [143] Formatted | Peter von Dassow | 11/29/17 8:45:00 AM |
|---|---|---|

Font:10 pt

| Page 21: [144] Formatted | Peter von Dassow | 11/29/17 10:38:00 AM |
|---|---|---|

Right, Right:  0 cm

| Page 21: [145] Formatted | Peter von Dassow | 11/29/17 8:45:00 AM |
|---|---|---|

Font:10 pt

| Page 21: [146] Formatted | Peter von Dassow | 11/29/17 8:45:00 AM |
|---|---|---|

Font:10 pt

| Page 21: [147] Formatted | Peter von Dassow | 11/29/17 8:45:00 AM |
|---|---|---|

Font:10 pt

| Page 21: [148] Formatted | Peter von Dassow | 11/29/17 8:45:00 AM |
|---|---|---|

Font:10 pt

| Page 21: [149] Formatted | Peter von Dassow | 11/29/17 8:45:00 AM |
|---|---|---|

Font:10 pt

| Page 21: [150] Formatted | Peter von Dassow | 11/29/17 10:08:00 AM |
|---|---|---|

Font:10 pt

| Page 21: [151] Formatted | Peter von Dassow | 11/29/17 10:08:00 AM |
|---|---|---|

Font:10 pt

| Page 21: [152] Formatted | Peter von Dassow | 11/29/17 10:37:00 AM |
|---|---|---|

Right, Right:  0.02 cm

| Page 21: [153] Formatted | Peter von Dassow | 11/29/17 8:45:00 AM |
|---|---|---|

Font:10 pt

| Page 21: [154] Formatted | Peter von Dassow | 11/29/17 10:37:00 AM |
|---|---|---|

Right, Right:  -0.01 cm

| Page 21: [155] Formatted | Peter von Dassow | 11/29/17 8:45:00 AM |
|---|---|---|

Font:10 pt

| Page 21: [156] Formatted | Peter von Dassow | 11/29/17 10:38:00 AM |
|---|---|---|

Right:  0 cm

| Page 21: [157] Formatted | Peter von Dassow | 11/29/17 8:45:00 AM |
|---|---|---|

Font:10 pt

| Page 21: [158] Formatted | Peter von Dassow | 11/29/17 10:38:00 AM |
|---|---|---|

Right, Right:  0 cm

| Page 21: [159] Formatted | Peter von Dassow | 11/29/17 8:45:00 AM |
|---|---|---|

Font:10 pt

| Page 21: [160] Formatted | Peter von Dassow | 11/29/17 8:45:00 AM |
|---|---|---|

Font:10 pt

| Page 21: [161] Formatted | Peter von Dassow | 11/29/17 8:45:00 AM |
|---|---|---|

Font:10 pt

| Page 21: [162] Formatted | Peter von Dassow | 11/29/17 8:45:00 AM |
|---|---|---|

Font:10 pt

| Page 21: [163] Formatted | Peter von Dassow | 11/29/17 8:45:00 AM |
|---|---|---|

Font:10 pt

| Page 21: [164] Formatted | Peter von Dassow | 11/29/17 10:08:00 AM |
|---|---|---|

Font:10 pt

| Page 21: [165] Formatted | Peter von Dassow | 11/29/17 10:08:00 AM |
|---|---|---|

Font:10 pt

| Page 21: [166] Formatted | Peter von Dassow | 11/29/17 10:03:00 AM |
|---|---|---|

Right

| Page 21: [167] Formatted | Peter von Dassow | 11/29/17 10:37:00 AM |
|---|---|---|

Indent: Left:  -0.35 cm, Right:  0.02 cm

| Page 21: [168] Formatted | Peter von Dassow | 11/29/17 10:37:00 AM |
|---|---|---|

Indent: Left:  -0.34 cm, Right:  -0.01 cm

| Page 21: [169] Formatted | Peter von Dassow | 11/29/17 10:38:00 AM |
|---|---|---|

Centered, Indent: Left:  -0.14 cm

| Page 21: [170] Formatted | Peter von Dassow | 11/29/17 10:38:00 AM |
|---|---|---|

Indent: Left:  -0.11 cm

| Page 21: [171] Formatted | Peter von Dassow | 11/29/17 10:38:00 AM |
|---|---|---|

Indent: Left:  -0.19 cm

| Page 21: [172] Formatted | Peter von Dassow | 11/29/17 10:39:00 AM |
|---|---|---|

Indent: Left:  -0.19 cm

| Page 21: [173] Formatted | Peter von Dassow | 11/29/17 10:39:00 AM |
|---|---|---|

Indent: Left:  -0.17 cm

| Page 21: [174] Formatted | Peter von Dassow | 11/29/17 10:39:00 AM |
|---|---|---|

Indent: Left:  -0.19 cm

| Page 21: [175] Formatted | Peter von Dassow | 11/29/17 10:40:00 AM |
|---|---|---|

Right, Right:  0 cm

| Page 21: [176] Formatted | Peter von Dassow | 11/29/17 10:08:00 AM |
|---|---|---|

Font:10 pt

**Page 21: [177] Formatted**      **Peter von Dassow**      **11/29/17 10:08:00 AM**

Font:10 pt

**Page 21: [178] Formatted**      **Peter von Dassow**      **11/29/17 10:37:00 AM**

Indent: Left:  -0.35 cm, Right:  0.02 cm

**Page 21: [179] Formatted**      **Peter von Dassow**      **11/29/17 10:37:00 AM**

Indent: Left:  -0.34 cm, Right:  -0.01 cm

**Page 21: [180] Formatted**      **Peter von Dassow**      **11/29/17 10:38:00 AM**

Centered, Indent: Left:  -0.14 cm

**Page 21: [181] Formatted**      **Peter von Dassow**      **11/29/17 10:38:00 AM**

Indent: Left:  -0.11 cm

**Page 21: [182] Formatted**      **Peter von Dassow**      **11/29/17 10:38:00 AM**

Indent: Left:  -0.19 cm

**Page 21: [183] Formatted**      **Peter von Dassow**      **11/29/17 10:39:00 AM**

Indent: Left:  -0.19 cm

**Page 21: [184] Formatted**      **Peter von Dassow**      **11/29/17 10:39:00 AM**

Indent: Left:  -0.17 cm

**Page 21: [185] Formatted**      **Peter von Dassow**      **11/29/17 10:39:00 AM**

Indent: Left:  -0.19 cm

**Page 21: [186] Formatted**      **Peter von Dassow**      **11/29/17 10:40:00 AM**

Right, Right:  0 cm

**Page 21: [187] Formatted**      **Peter von Dassow**      **11/29/17 10:08:00 AM**

Font:10 pt

**Page 21: [188] Formatted**      **Peter von Dassow**      **11/29/17 10:08:00 AM**

Font:10 pt

**Page 21: [189] Formatted**      **Peter von Dassow**      **11/29/17 10:03:00 AM**

Right

**Page 21: [190] Formatted**      **Peter von Dassow**      **11/29/17 10:37:00 AM**

Indent: Left:  -0.35 cm, Right:  0.02 cm

**Page 21: [191] Formatted**      **Peter von Dassow**      **11/29/17 10:37:00 AM**

Indent: Left:  -0.34 cm, Right:  -0.01 cm

**Page 21: [192] Formatted**      **Peter von Dassow**      **11/29/17 10:38:00 AM**

Centered, Indent: Left:  -0.14 cm

| Page 21: [193] Formatted | Peter von Dassow | 11/29/17 10:38:00 AM |
|---|---|---|

Indent: Left: -0.11 cm

| Page 21: [194] Formatted | Peter von Dassow | 11/29/17 10:38:00 AM |
|---|---|---|

Indent: Left: -0.19 cm

| Page 21: [195] Formatted | Peter von Dassow | 11/29/17 10:39:00 AM |
|---|---|---|

Indent: Left: -0.19 cm

| Page 21: [196] Formatted | Peter von Dassow | 11/29/17 10:39:00 AM |
|---|---|---|

Indent: Left: -0.17 cm

| Page 21: [197] Formatted | Peter von Dassow | 11/29/17 10:39:00 AM |
|---|---|---|

Indent: Left: -0.19 cm

| Page 21: [198] Formatted | Peter von Dassow | 11/29/17 10:40:00 AM |
|---|---|---|

Right, Right: 0 cm

| Page 21: [199] Formatted | Peter von Dassow | 11/29/17 10:08:00 AM |
|---|---|---|

Font:10 pt

| Page 21: [200] Formatted | Peter von Dassow | 11/29/17 10:08:00 AM |
|---|---|---|

Font:10 pt

| Page 21: [201] Formatted | Peter von Dassow | 11/29/17 10:37:00 AM |
|---|---|---|

Indent: Left: -0.35 cm, Right: 0.02 cm

| Page 21: [202] Formatted | Peter von Dassow | 11/29/17 10:37:00 AM |
|---|---|---|

Indent: Left: -0.34 cm, Right: -0.01 cm

| Page 21: [203] Formatted | Peter von Dassow | 11/29/17 10:38:00 AM |
|---|---|---|

Centered, Indent: Left: -0.14 cm

| Page 21: [204] Formatted | Peter von Dassow | 11/29/17 10:38:00 AM |
|---|---|---|

Indent: Left: -0.11 cm

| Page 21: [205] Formatted | Peter von Dassow | 11/29/17 10:38:00 AM |
|---|---|---|

Indent: Left: -0.19 cm

| Page 21: [206] Formatted | Peter von Dassow | 11/29/17 10:39:00 AM |
|---|---|---|

Indent: Left: -0.19 cm

| Page 21: [207] Formatted | Peter von Dassow | 11/29/17 10:39:00 AM |
|---|---|---|

Indent: Left: -0.17 cm

| Page 21: [208] Formatted | Peter von Dassow | 11/29/17 10:39:00 AM |
|---|---|---|

Indent: Left: -0.19 cm

| **Page 21: [209] Formatted** | **Peter von Dassow** | **11/29/17 10:40:00 AM** |

Right, Right:  0 cm

| **Page 21: [210] Formatted** | **Peter von Dassow** | **11/29/17 10:08:00 AM** |

Font:10 pt

| **Page 21: [211] Formatted** | **Peter von Dassow** | **11/29/17 10:08:00 AM** |

Font:10 pt

| **Page 21: [212] Formatted** | **Peter von Dassow** | **1/22/18 6:21:00 PM** |

Font:Italic

| **Page 24: [213] Deleted** | **Peter von Dassow** | **1/25/18 9:00:00 AM** |